# Interpretable Unsupervised Joint Denoising and Enhancement for Real-World low-light Scenarios

**Huaqiu Li, Xiaowan Hu, Haoqian Wang** *
Tsinghua Shenzhen International Graduate School
Tsinghua University
`{lihq23, hu-xw19}@mails.tsinghua.edu.cn`

## Abstract

Real-world low-light images often suffer from complex degradations such as local overexposure, low brightness, noise, and uneven illumination. Supervised methods tend to overfit to specific scenarios, while unsupervised methods, though better at generalization, struggle to model these degradations due to the lack of reference images. To address this issue, we propose an interpretable, zero-reference joint denoising and low-light enhancement framework tailored for real-world scenarios. Our method derives a training strategy based on paired sub-images with varying illumination and noise levels, grounded in physical imaging principles and retinex theory. Additionally, we leverage the Discrete Cosine Transform (DCT) to perform frequency domain decomposition in the sRGB space, and introduce an implicit-guided hybrid representation strategy that effectively separates intricate compounded degradations. In the backbone network design, we develop retinal decomposition network guided by implicit degradation representation mechanisms. Extensive experiments demonstrate the superiority of our method. Code will be available at `https://github.com/huaqlili/ unsupervised-light-enhance-ICLR2025`.

## 1 INTRODUCTION

Low-light image enhancement is a significant research area in computer vision and image processing. The inherently low signal-to-noise ratio of such images can adversely impact downstream tasks, such as object detection Rashed et al. (2019), image segmentation Wang et al. (2022), and face recognition Serengil & Ozpinar (2020). Moreover, the widespread application of low-light enhancement in fields like nighttime photography Jin et al. (2022; 2023), astronomical observation Chen et al. (2021), and autonomous driving Li et al. (2024) underscores its critical importance in low-level vision tasks.

Real-world low-light enhancement presents numerous challenges, requiring simultaneous handling of issues such as brightness, contrast, artifacts, and noise. Over the past few decades, traditional methods like gamma correction Huang et al. (2012), histogram equalization Lee et al. (2013), and retinex theory Land & McCann (1971) have been developed. However, these methods focus on single-dimensional brightness issues and struggle with complex real-world scenes, while their hand-crafted priors often lack generalization for diverse conditions.

In recent years, learning-based methods for low-light enhancement have achieved significant progress. However, these approaches often rely on paired (e.g. Zhang et al. (2021); Cai et al. (2023); Wu et al. (2022); Xing et al. (2024); Bai et al. (2024)) or unpaired (e.g. Jiang et al. (2021); Yang et al. (2023)) data, making it challenging to collect large-scale datasets. Additionally, discrepancies in brightness between reference images can disrupt model fitting, making the development of efficient zero-reference methods crucial.

Current zero-reference low-light enhancement methods, such as Zero-DCE Guo et al. (2020), utilize curve learning for iterative optimization, but does not account for noise degradation. Approaches like SCI Ma et al. (2022) and RUAS Liu et al. (2021) follow a similar iterative strategy, integrating denoising modules. However, while separate denoising modules are designed for end-to-end

---
*Corresponding author.

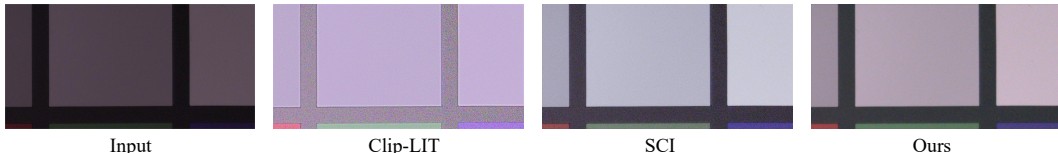

| Input | Clip-LIT | SCI | Ours |

Figure 1: Compared with state-of-the-art methods Liang et al. (2023); Ma et al. (2022) on the SIDD dataset, our approach achieves the best results in denoising, enhancement, and color fidelity grounded in real-world imaging principles.

training, they rely on specific, lengthy loss functions that lack generalization across various noise patterns. Other methods Fan et al. (2022) address multiple degradation tasks through multi-stage learning. They often overlook the error accumulation during the optimization process (e.g., noise becomes more complex after low-light enhancement). Furthermore, as Fig.1 shows, these methods generally fail to differentiate feature layers for multiple degradation modes, leading to confusion and ambiguity during the restoration process.

To address the aforementioned challenges, we propose a zero-reference joint denoising and enhancement method grounded in real-world physical models. Specifically, we introduce a self-supervised image denoising method based on neighboring pixel masking, alongside a self-supervised enhancement strategy that combines random gamma adjustment with retinex theory. By obtaining sub-image pairs with varying illumination and noise levels, the framework is capable of tackling the complex degradation issues caused by low-light conditions. Additionally, we employed DCT to model physical priors that reflect various degradations, and designed a global learning-based encoder to extract implicit degradation representations from them. In the backbone network design, we develop retinal decomposition network guided by implicit degradation representation mechanisms. This approach allows us to separate and address complex degradations in the frequency domain, rather than sequentially handling features as in previous methods. Extensive experiments demonstrate that our method offers significant advantages over the current SOTA approaches.

The main contributions of this paper are as follows:

- By preprocessing the original low-light image to generate paired sub-images with varying illumination and noise levels, followed by retinal decomposition, we derived and validated a physically sound unsupervised joint denoising and enhancement framework.

- We utilized DCT to model physical priors that capture intricate compounded degradations, and designed a globally learned encoder to extract implicit degradation representations from these priors.

- We developed a hybrid-prior attention transformer network that integrates degradation features to reconstruct the reflection map, while adaptively enhancing the illumination.

- Extensive experiments on multiple real-world datasets demonstrate that our method achieves superior performance across several metrics compared to SOTA approaches.

## 2 RELATED WORKS

### 2.1 SELF-SUPERVISED/UNSUPERVISED LOW-LIGHT IMAGE ENHANCEMENT

The development of self-supervised and unsupervised low-light enhancement follows two main approaches: zero-reference and unpaired learning. Zero-DCE Guo et al. (2020) introduced a curve-based iterative method for zero-reference enhancement, later refined by Zero-DCE++ Li et al. (2021) for better efficiency. Methods like RUAS Liu et al. (2021) and SCI Ma et al. (2022) extend this approach with denoising modules for handling complex degradations. However, these approaches often struggle with interpretability and modeling complex degradations. In contrast, unpaired learning leverages low-light and normal-light image pairs from different scenes or varying illumination within the same scene. GAN-based methods like EnlightenGAN Jiang et al. (2021) and NeRCo Yang et al. (2023) use cyclical networks for bidirectional image transformation learning between domains. PairLIE Fu et al. (2023) processes low-light images with varying degradations from the same scene using retinal theory. Although these methods demonstrate strong generative abilities, their performance can be constrained by inconsistent normal-light references and difficulties in normalizing illumination distributions.

## 2.2 FREQUENCY-DOMAIN ANALYSIS IN IMAGE PROCESSING

DCTconv Chęiński & Wawrzyński (2020) integrates convolution with IDCT to form a novel layer that facilitates network pruning. Xie et al. (2021) introduces a frequency-aware dynamic network that leverages DCT in image super-resolution to reduce computational cost. To improve content preservation, Cai et al. (2021) employs a Fourier frequency spectrum consistency constraint for image translation. Recently, frequency domain processing has gained significant attention. Zou et al. (2024) demonstrates that degradation predominantly affects amplitude spectra, while FSI Liu et al. (2023) designs a frequency-spatial interactive network to address under-display camera image restoration. Zou et al. (2022) employs wavelet transforms to disentangle frequency domain information, using a multi-branch network to recover high-frequency details. WINNet Ou et al. (2024) combines wavelet-based and learning-based methods to construct a reversible, interpretable network with strong generalization capabilities. FCDiffusion Gao et al. (2024) utilizes DCT to filter feature maps, achieving controlled generation across different frequency bands.

## 3 METHOD

### 3.1 THEORETICAL BASIS

#### 3.1.1 RETINEX THEORY

The traditional Retinex image enhancement algorithm Land & McCann (1971); Wei et al. (2018) simulates human visual perception of brightness and color. It decomposes image $I \in \mathbb{R}^{H \times W \times 3}$ into the illumination component $L \in \mathbb{R}^{H \times W \times 3}$ and the reflection component $R \in \mathbb{R}^{H \times W \times 3}$. This conclusion can be expressed by the following formula:

$$I = R \circ L \qquad (1)$$

where $\circ$ denotes element-wise multiplication. The reflection component $R$ is determined by the intrinsic properties of the object, while the illumination component $L$ represents the lighting intensity. However, the traditional Retinex algorithm does not account for complex degeneration produced by unbalanced light distribution or real-world dark scenes in low-light conditions, and this loss of quality is further amplified with the enhancement of the image. Therefore, we add the noise disturbance term $N$ on the reflection component as the basis of theoretical analysis:

$$I = (R + N) \circ L \qquad (2)$$

In most low-light scenarios, $N$ is modeled as zero-mean Poisson noise.

#### 3.1.2 NEIGHBORING PIXEL MASKING IN SELF-SUPERVISED DENOISING

Image denoising represents a classic ill-posed problem within the domain of image restoration. This signifies the existence of multiple potential solutions for the same noisy scene. Previous image denoising models Zhang et al. (2017); Goyal et al. (2020) typically require paired input of noisy images $\mathbf{y}_i$ and corresponding clean images $\mathbf{x}_i$ to train the network effectively.

$$\arg \min_{\theta} \sum_i L(f_\theta(\mathbf{y}_i), \mathbf{x}_i) \qquad (3)$$

Here, $\theta$ represents parameters that need to be optimized. However, in practical scenarios, obtaining paired images is often challenging or even impossible. As a result, a series of self-supervised and unsupervised methods have emerged utilizing only noisy images for training.

The theoretical foundation of N2NLehtinen et al. (2018) is rooted in point estimation, which estimates the true value of a series of observations $\{\mathbf{x}_1, \mathbf{x}_2, ..., \mathbf{x}_n\}$. The objective is to find a value $\mathbf{z}$ that minimizes the sum of distances to all the observed values, serving as the estimation. When using $\mathcal{L}_2$ loss for estimation, replacing $x$ with another observation $\mathbf{z}$ having the same mean value does not alter the result.

Extending this theoretical point estimation framework to training neural network regressors, the optimization objective of the network can be transformed into:

$$\arg \min_{\theta} \sum_i L(f_\theta(\mathbf{y}_i), \mathbf{z}) \qquad (4)$$

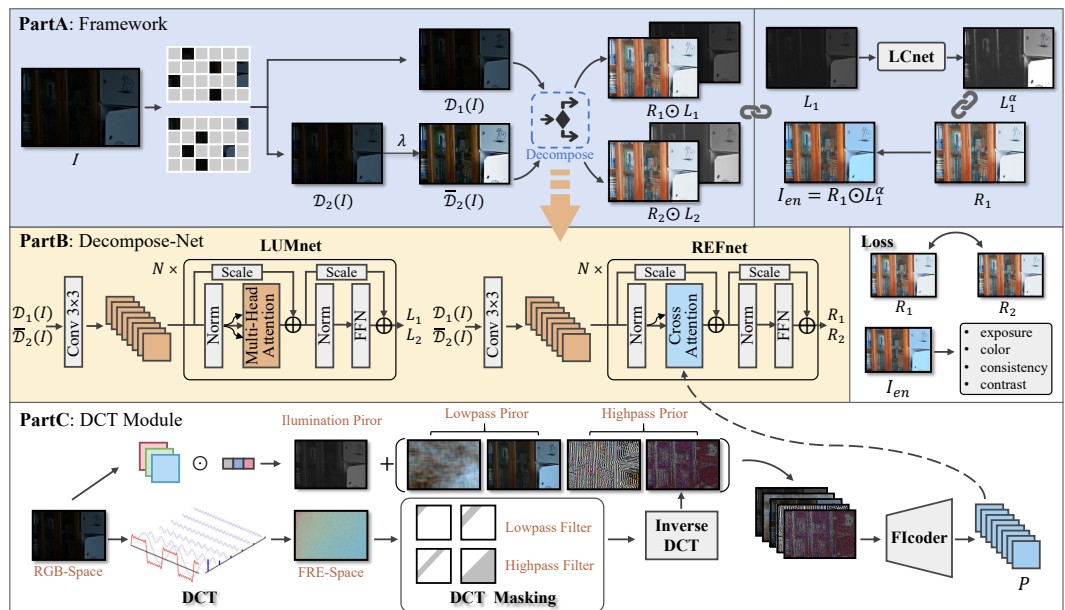

Figure 2: The pipeline of our proposed method: First, we preprocess the low-light full-resolution image $I$ using pixel masks and gamma-based nonlinear enhancement, generating sub-images with varying illumination and noise levels. These are then processed through Decompose-Net, which uses a transformer architecture integrating hybrid degradation representations, incorporating cross-attention to inject guiding embeddings. Subsequently, LCnet enhances the illumination map.

This implies that when training a denoising network, if we replace the clean images $\mathbf{x}_i$ with noisy images $\mathbf{z}$, which have zero-mean noise, the optimization results using L2 loss will be equivalent to those trained by pairs of noisy-clean images. This assumption forms the foundation of our work.

## 3.2 OVERALL ARCHITECTURE

Building on the aforementioned theoretical foundation, we express a low-light image $I = (R + N) \circ L$, where $N$ represents a zero-mean noise distribution. Our objective is to generate images of the same scene with differing noise observations, ensuring that the noise remains zero-mean and the denoised ground truth is consistent across these images. In scenarios where a normal-light reference image is unavailable, we propose to generate two sub-images at 1/4 resolution through a process of neighboring masking $\mathcal{D}$. Specifically, the original image $I$ is partitioned into multiple 2x2 pixel patches. From each patch, two adjacent pixels are randomly selected and assigned to corresponding regions in the two sub-images. The resulting sub-images can thus be mathematically formulated as:

$$\mathcal{D}_1(I) = (R_1 + N_1) \circ L_1, \mathcal{D}_2(I) = (R_2 + N_2) \circ L_2 \tag{5}$$

Here, $N_1$ and $N_2$ represent noise components that follow a shared distribution, $R_1$ and $R_2$ are highly similar in pixel values, and $L_1$ and $L_2$ correspond to the same lighting conditions.

Previous study Fu et al. (2023) has indicated that if images of the same scene under different illumination conditions can be obtained, deep learning can be employed to decompose the corresponding reflectance $R$, with the principle that the reflectance $R_1$ and $R_2$ should theoretically be identical. To generate a supervision signal with different illumination, we apply gamma correction to $\mathcal{D}_2(I)$ and get $\overline{\mathcal{D}}_2(I)$. We avoid applying gamma correction directly to the original image $I$ because the noise $N$ would be preserved at nearly the same level, making the network learn an identity mapping. After obtaining the enhanced image $\overline{\mathcal{D}}_2(I)$, and given that $N_2$ is relatively small compared to the pixel values, we further perform a Taylor series expansion on it:

$$\overline{\mathcal{D}}_2(I) = \mathcal{D}_2(I)^\lambda = (R_2 + N_2)^\lambda \circ L_2^\lambda \approx (R_2^\lambda + \lambda R_2^{\lambda-1} N_2) \circ L_2^\lambda = (R_2 + \lambda N_2) \circ R_2^{\lambda-1} \circ L_2^\lambda \tag{6}$$

Here, $\lambda$ represents the gamma enhancement factor, and $R^{\lambda-1} \approx 1$ when $\lambda$ is close to 1. The original equation can thus be further rewritten as $(R_2 + \lambda N_2) \circ \overline{L}_2, \overline{L}_2 = L_2^\lambda$, leading to the final expressions

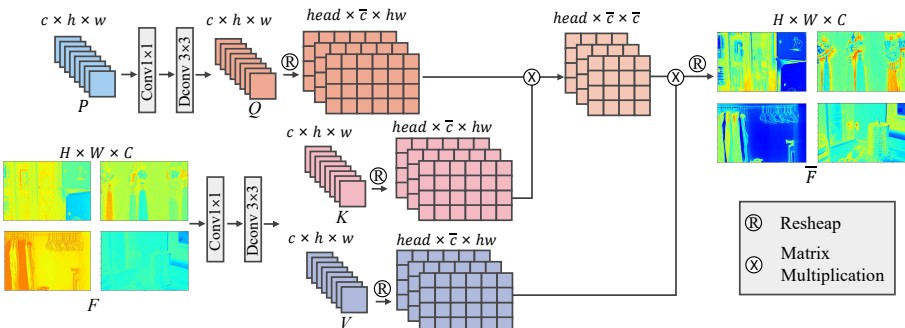

Figure 3: Illustration of the hybrid prior degradation representation guided by multi-head cross attention. After processing, the feature maps exhibit clearer hierarchical structure and reduced noise.

for the two sub-images:

$$\mathcal{D}_1(I) = (R_1 + N_1) \circ L_1, \overline{\mathcal{D}}_2(I) = (R_2 + \lambda N_2) \circ \overline{L}_2 \tag{7}$$

In this formulation, $R_1$ and $R_2$ share the same ground truth reflectance, as they exist within the same scene. Meanwhile, $N_1$ and $\lambda N_2$ represent zero-mean noise distributions that are non-identical. Additionally, the first and second images encompass different illumination conditions. Therefore, by simply constraining $(R_1 + N_1)$ and $(R_2 + \lambda N_2)$ to be equal, we can construct a self-supervised network jointly performing denoising and enhancement (DEnet).

As illustrated in Fig.2, the overall architecture of DEnet is primarily divided into four components: the Frequency-Illumination representation Encoder (FIcoder), the Reflectance Map Extraction Network (REFnet), the Illumination Map Extraction Network (LUMnet), and the Light Correction Network (LCnet). REFnet and LUMnet are employed to extract the reflectance maps $R_1$, $R_2$, and illumination maps $L_1$, $\overline{L}_2$ from sub-images $\mathcal{D}_1(I)$ and $\overline{\mathcal{D}}_2(I)$. In LUMnet, each transformer block is divided into a self-attention computation module and a gating module. In contrast, REFnet, tasked with reflectance map extraction, requires the degradation representations to perform cross-attention calculations with feature tokens ss illustrated in the Fig.3. LCnet processes its features using a transformer and then applies global average pooling. The pooled features are passed through two linear layers to scale them into a one-dimensional enhancement factor to correct the illumination map, which is subsequently multiplied with the reflectance map to produce the final corrected image $I_{en}$.

### 3.3 FREQUENCY-ILLUMINATION PRIOR ENCODER

FIcoder is primarily designed to obtain degradation representations from illumination and frequency domain priors, which are then integrated with feature maps through cross-attention mechanisms in REFnet. The fusion of multiple priors enhances the model's generalization capability across diverse and complex degradations. As illustrated in the Fig.4, the illumination prior $I_{lu}$ represents the image's luminance information, while the four frequency domain priors $C_{low\_1}, C_{low\_2}, C_{high\_1}, C_{high\_2}$, ranging from low to high frequencies, capture information on chromaticity, semantics, edge contours, and noise intensity, respectively.

First, we extract the illumination prior $I_{lu} = mean_c(I)$, which is the mean value of the sub-image across the channel dimension, representing the overall brightness level of the image. As for frequency prior, we use channel-wise 2D DCT to convert the spatial-domain image $I$ into the frequency-domain counterpart $F$. Different spectral bands in the DCT domain encode different image visual attributes degradation representation analysis of input images. To obtain the frequency spectrum maps across four frequency bands, we define four masks:

$$M_{low\_1}(u,v) = 1 \ if \ u+v \le t \ else \ 0, M_{low\_2}(u,v) = 1 \ if \ u+v \le 3t \ else \ 0, \tag{8}$$

$$M_{high\_1}(u,v) = 1 \ if \ 2t < u+v \le 4t \ else \ 0, M_{high\_2}(u,v) = 1 \ if \ u+v \ge 5t \ else \ 0. \tag{9}$$

$$F^* = F \times M^*, \tag{10}$$

where $* \in \{low\_1, low\_2, high\_1, high\_2\}$, and $t$ represents the manually set bandwidth hyperparameter. We apply the masks $M_*$ to the frequency spectrum feature maps $F$ to filter them according to different frequency bands. By performing an inverse Discrete Cosine Transform (IDCT) on these filtered maps $F_*$, we obtain the corresponding spatial domain feature images $C_*$.

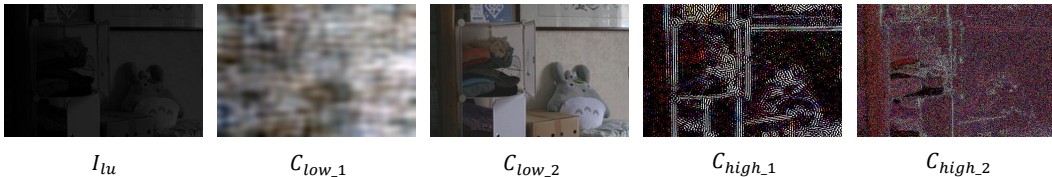

$$I_{lu} \qquad C_{low\_1} \qquad C_{low\_2} \qquad C_{high\_1} \qquad C_{high\_2}$$

Figure 4: The visualization of the five image priors. They represent chromaticity, semantic information, edge contours, and noise intensity.

Finally, we combine $I_{lu} \in \mathbb{R}^{H \times W \times 1}$, $C_{low\_1}, C_{low\_2}, C_{high\_1}$ and $C_{high\_2} \in \mathbb{R}^{H \times W \times 3}$ through a convolutional network-based Illumination-Frequency Prior Encoder. This encoder constructs the implicit representation $P \in \mathbb{R}^{H \times W \times C}$, based on separating degradation features. During training, the FIcoder processes the input sub-images $\mathcal{D}_1(I)$ and $\overline{\mathcal{D}}_2(I)$, generating the corresponding degradation representations $P_1$ and $P_2$.

### 3.4 LOSS FUNCTION

During model training, DEnet performs the following computations:

$$I_{en} = DE(\mathcal{D}_1(I)) = R_1 \circ L_1^\alpha, R_1 = REF(\mathcal{D}_1(I), P_1), L_1 = LUM(\mathcal{D}_1(I)), \alpha = LC(L_1) \quad (11)$$

During inference, we input the original-resolution low-light image $I$, multiply the decomposed reflection component $R$ with the corrected illumination $L$, and obtain the final enhanced result.

The loss function for this method is primarily divided into two aspects: **1) Retinex Decomposition Loss:** This loss constrains the retinex decomposition to ensure that the resulting reflectance and illumination maps are consistent with the underlying physical assumptions. **2) Self-supervised Enhancement Loss:** This loss is designed to regulate the enhanced image $I_{en}$ by imposing constraints on brightness, contrast, saturation, and other factors, ensuring that the enhancement aligns with desired visual qualities.

The Retinal Decomposition Loss we employ is primarily divided into two parts: the first is $\mathcal{L}_R$, as mentioned earlier, which primarily constrains the $L_2$ distance between the reflectance maps $R_1$ and $R_2$ derived from $\mathcal{D}_1(I)$ and $\overline{\mathcal{D}}_2(I)$; the second is $\mathcal{L}_L$, which imposes smoothness constraints on the illumination maps and ensures that the product of the decomposed maps equals the original image. The expressions for these two losses are shown as follows:

$$\mathcal{L}_R = \left\| REF(\mathcal{D}_1(I), P_1) - REF(\overline{\mathcal{D}}_2(I), P_2) \right\|_2^2 + \omega_{reg} \mathcal{L}_{reg} \quad (12)$$

$$\mathcal{L}_L = \| R_1 \circ L_1 - \mathcal{D}_1(I) \|_2^2 + \| L_1 - L_0 \|_2^2 + \left\| R_1 - \frac{\mathcal{D}_1(I)}{L_1.detach()} \right\|_2^2 + \bigtriangledown L_1, L_0 = \max_{c \in \{r,g,b\}} \mathcal{D}_1(I)_c \quad (13)$$

Here, $P_1$ and $P_2$ represent the degradation representations extracted by the FIcoder from the sub-images $\mathcal{D}_1(I)$ and $\overline{\mathcal{D}}_2(I)$, respectively. $\bigtriangledown L_1$ denotes the gradient of the illumination map. We add a regularization term $\mathcal{L}_{reg}$ to align gradients and test the original-scale images. The masked testing results are compared with sub-image reflectance maps via $L_2$-norm, ensuring the consistency of $R_1$ and $R_2$ across scales, enhancing generalization and training stability.

$$\mathcal{L}_{reg} = \left\| REF(\mathcal{D}_1(I), P_1) - REF(\overline{\mathcal{D}}_2(I), P_2) - (\mathcal{D}_1(REF(I, P)) - \overline{\mathcal{D}}_2(REF(I, P))) \right\|_2^2 \quad (14)$$

For the Self-supervised Enhancement Loss, we designed two components: the consistency loss $L_{con}$ and the enhancement loss $L_{enh}$:

$$\mathcal{L}_{con} = \frac{1}{K} \sum_{i=1}^{K} \sum_{j \in \sigma(i)} (|I_{en,i} - I_{en,j}| - |\mathcal{D}_1(I)_i - \mathcal{D}_1(I)_j|) \quad (15)$$

$$\mathcal{L}_{enh} = \omega_{exp} \frac{1}{K} \sum_{i=1}^{K} |I_{en,i} - E| + \omega_{col} \sum_{\forall (p,q) \in \varepsilon} (V_p - V_q)^2, \varepsilon = \{(R,G), (R,B), (G,B)\} \quad (16)$$

The images before and after enhancement are divided into $K$ patches. Here, $\sigma(i)$ represents the neighboring patches surrounding position $i$. $I_{en,i}$ and $\mathcal{D}_1(I)_i$ denote the mean pixel values within

the i-th patches at the corresponding position. The loss $L_{enh}$ imposes constraints on the average brightness of the patches and the overall chromaticity of the image, where $V_p$ denotes the average intensity value of $p$ channel in the enhanced image, and $E$ represents the exposure standard that aligns with natural perception. $\omega_{exp}$ and $\omega_{col}$ represent the respective weighting factors. Finally, the overall loss of the end-to-end network can be described as follows:

$$\mathcal{L} = \omega_R \mathcal{L}_R + \omega_L \mathcal{L}_L + \omega_{con} \mathcal{L}_{con} + \omega_{enh} \mathcal{L}_{enh} \tag{17}$$

Here, $\omega_R$, $\omega_L$, $\omega_{con}$, and $\omega_{enh}$ represent the respective weighting factors.

Table 1: PSNR↑, SSIM↑, LPIPS↓ scores on the image sets (LOLv1, LOLv2). The best result is in red, whereas the second-best one is in blue under each case.

| Method | Reference | LOLv1 | | | LOLv2-Real | | |
|---|---|---|---|---|---|---|---|
| | | PSNR↑ | SSIM↑ | LPIPS↓ | PSNR↑ | SSIM↑ | LPIPS↓ |
| **Supervised** | | | | | | | |
| URetinexNet | Wu et al. (2022) | 19.84 | 0.824 | 0.237 | 21.09 | 0.858 | 0.208 |
| SNR-aware | Xu et al. (2022) | 24.61 | 0.842 | 0.233 | 21.48 | 0.849 | 0.237 |
| LLFormer | Wang et al. (2023) | 23.65 | 0.818 | 0.169 | 27.75 | 0.861 | 0.142 |
| Retinexformer | Cai et al. (2023) | 23.93 | 0.831 | —— | 21.23 | 0.838 | —— |
| Retinexmamba | Bai et al. (2024) | 24.03 | 0.831 | —— | 22.45 | 0.844 | —— |
| **Unpaired** | | | | | | | |
| EnlightenGAN | Jiang et al. (2021) | 17.48 | 0.651 | 0.322 | 18.64 | 0.675 | 0.308 |
| PairLIE | Fu et al. (2023) | 19.51 | 0.736 | **0.247** | **19.70** | **0.774** | **0.235** |
| Nerco | Yang et al. (2023) | 19.70 | **0.742** | **0.234** | 19.66 | 0.717 | 0.270 |
| **No-Reference** | | | | | | | |
| ZERO-DCE | Guo et al. (2020) | 14.86 | 0.559 | 0.335 | 18.06 | 0.573 | 0.312 |
| RUAS | Liu et al. (2021) | 16.40 | 0.500 | 0.270 | 15.33 | 0.488 | 0.310 |
| Sci-easy | Ma et al. (2022) | 9.58 | 0.369 | 0.410 | 11.98 | 0.399 | 0.354 |
| Sci-medium | | 14.78 | 0.522 | 0.339 | 17.30 | 0.534 | 0.308 |
| Sci-hard | | 13.81 | 0.526 | 0.358 | 17.25 | 0.546 | 0.317 |
| Clip-LIT | Liang et al. (2023) | 17.21 | 0.589 | 0.335 | 17.06 | 0.589 | 0.352 |
| Enlighten-Your-Voice | Zhang et al. (2023) | **19.73** | 0.715 | —— | 19.34 | 0.686 | —— |
| Ours | | **19.80** | **0.750** | 0.253 | **20.22** | **0.793** | **0.266** |

Table 2: PSNR↑/SSIM↑/LPIPS↓ scores on the image set SICE, and BRSIQUE↓/CLIPIQA↓ scores on the image set SIDD. The best result is in red, whereas the second-best one is in blue.

| Method | Parameters | SICE | | | SIDD | |
|---|---|---|---|---|---|---|
| | | PSNR↑ | SSIM↑ | LPIPS↓ | BRSIQUE↓ | CLIPIQA↓ |
| **Supervised** | | | | | | |
| URetinexNet | 1.04M | 22.12 | 0.844 | 0.462 | —— | —— |
| SNR-aware | 50.95M | 15.02 | 0.584 | 0.527 | 25.679 | 0.294 |
| LLFormer | 72.29M | 17.88 | 0.821 | 0.503 | 3.548 | 0.339 |
| Retinexformer | 1.61M | —— | —— | —— | 9.229 | 0.343 |
| Retinexmamba | 4.59M | —— | —— | —— | 11.826 | 0.386 |
| **Unpaired** | | | | | | |
| EnlightenGAN | 8.44M | 18.73 | 0.822 | **0.216** | 13.786 | **0.337** |
| PairLIE | 0.34M | **21.32** | **0.840** | **0.216** | **3.168** | 0.383 |
| Nerco | 22.76M | 18.72 | 0.805 | 0.474 | —— | —— |
| **No-Reference** | | | | | | |
| ZERO-DCE | 0.08M | 18.69 | 0.810 | 0.279 | 24.291 | 0.503 |
| RUAS | 0.01M | 13.18 | 0.734 | 0.363 | 31.613 | 0.361 |
| Sci-easy | 0.01M | 11.71 | 0.590 | 0.502 | 25.344 | 0.399 |
| Sci-medium | | 15.95 | 0.787 | 0.335 | 21.636 | 0.456 |
| Sci-hard | | 17.59 | 0.782 | 0.486 | 35.533 | 0.508 |
| Clip-LIT | 0.27M | 13.70 | 0.725 | 0.480 | 31.093 | 0.434 |
| Ours | 0.36M | **22.55** | **0.841** | **0.234** | **2.555** | **0.292** |

## 4 EXPERIMENT

### 4.1 IMPLEMENTATION DETAILS

To ensure fairness, all experiments were terminated after 100 training epochs. We consistently set the initial learning rate to $1 \times 10^{-5}$ and conducted all experiments on an RTX 3090 GPU. During training, images were randomly cropped into 256x256 patches, with pixel values normalized to the range of (0, 1), and a batch size of 1 was employed.

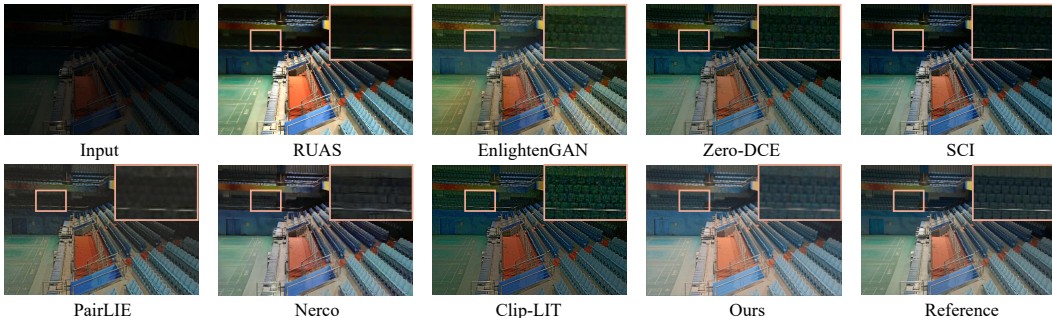

Figure 5: Visual comparison of typical unsupervised enhancement methods in LOL Yang et al. (2021). Flesh pink boxes indicate the obvious differences.

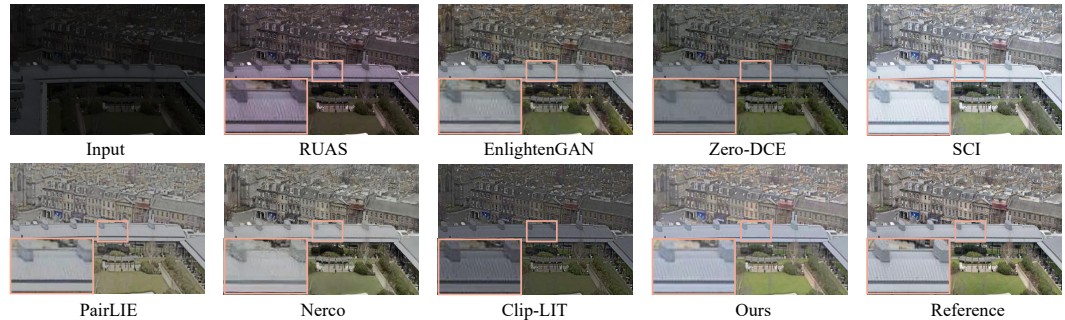

Figure 6: Visual comparison of typical unsupervised enhancement methods in SICE Cai et al. (2018). Flesh pink boxes indicate the obvious differences.

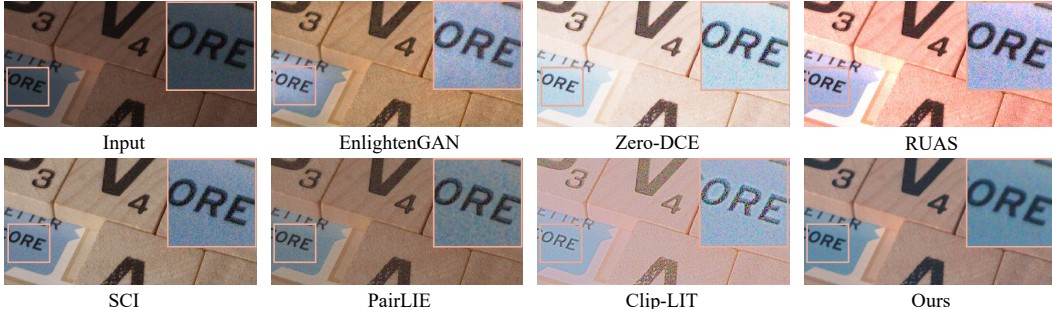

Figure 7: Visual comparison on the real-world low-light image from the SIDD Abdelhamed et al. (2018) dataset.

We conducted tests on four benchmarks: LOLv1 Wei et al. (2018), LOLv2-real Yang et al. (2021), SICE Cai et al. (2018) and SIDD Abdelhamed et al. (2018). Please refer to the supplementary materials for detailed information regarding the datasets, including the corresponding training and testing splits.

### 4.2 BENCHMARKING RESULTS

The experimental results on the LOL dataset are presented in Tab. 1, where our model outperforms most of the compared unpaired and no-reference methods, achieving the highest scores across multiple metrics. The qualitative visual comparisons are shown in Fig. 5. Unpaired methods benefit from reference images captured in normal lighting conditions, making learning the necessary illumination features easier. However, these methods struggle with underexposed local regions, leading to issues like dead black spots.

Meanwhile, EnlightGAN, ZeroDCE, and Clip-LIT successfully enhance overly dark regions. However, due to the lack of proper denoising mechanisms, they tend to introduce noise while increasing exposure. Our approach, leveraging illumination priors and frequency domain decomposition, effectively compensates for multidimensional illumination information, resolving complex degradation issues such as local overexposure, underexposure, and noise.

The experimental results on the SICE and SIDD datasets are shown in Tab. 2. The selected SICE test set includes images with three levels of low-light degradation: low, medium, and high. We evaluate the generalization capability of our model under varying illumination conditions using statistical metrics, and the qualitative comparisons are shown in Fig. 6. Both RUAS and EnlightenGAN exhibit issues such as local overexposure and strong contrast distortion, which can be attributed to the lack of an interpretable illumination feedback design in their network structures. Nerco generates artifacts in certain image regions, highlighting the uncontrollability of generative models in image enhancement tasks. In contrast, our method demonstrates appropriate contrast, accurate chrominance, low noise, and sufficient detail.

The qualitative comparison results on the SIDD dataset are shown in Fig. 7. We assess the enhancement capability of our model in challenging low-light scenes with high noise levels and complex noise patterns. Our method achieves the best performance on two no-reference statistical metrics, BRISQUE and CLIPIQA, indicating that the enhanced images exhibit characteristics closer to natural images with fewer distortions. From the visual results, our method demonstrates robustness against complex noise in real-world scenarios, effectively enhancing image illumination while controlling noise intensity. In contrast, other approaches either lack a dedicated denoising design or handle noise from a perceptual standpoint, without corresponding theoretical analysis for interpretability, leading to suboptimal results.

Table 3: Ablation study of the contribution of the three physical priors. The best and the second best results are highlighted in red and blue.

| | | | LOLv1 | | | LOLv2 | | |
|---|---|---|---|---|---|---|---|---|
| Illumination | Lowpass | Highpass | PSNR↑ | SSIM↑ | LPIPS↓ | PSNR↑ | SSIM↑ | LPIPS↓ |
| × | × | × | 18.88 | 0.741 | 0.273 | 19.37 | 0.771 | 0.305 |
| ✓ | × | × | 19.54 | **0.753** | **0.253** | **19.99** | **0.785** | 0.297 |
| ✓ | × | ✓ | **19.69** | 0.744 | **0.259** | 19.28 | 0.779 | 0.299 |
| ✓ | ✓ | × | 19.57 | 0.745 | 0.262 | 19.51 | 0.780 | **0.282** |
| ✓ | ✓ | ✓ | **19.80** | **0.750** | **0.253** | **20.22** | **0.793** | **0.266** |

Table 4: Ablation study of the contribution of the denoising designs, where NM stands for neighborhood masking. The best and the second best results are highlighted in red and blue.

| | | | LOLv1 | | | LOLv2 | | |
|---|---|---|---|---|---|---|---|---|
| Setting | NM | $\mathcal{L}_{reg}$ | PSNR↑ | SSIM↑ | LPIPS↓ | PSNR↑ | SSIM↑ | LPIPS↓ |
| 1 | × | × | 18.52 | 0.686 | 0.271 | 19.46 | 0.771 | 0.323 |
| 2 | ✓ | × | 19.63 | 0.747 | 0.264 | 19.83 | 0.787 | 0.279 |
| 3 | ✓ | ✓ | **19.80** | **0.750** | **0.253** | **20.22** | **0.793** | **0.266** |

## 4.3 ABLATION STUDY

**Denoiseing Design.** In the previous sections, we designed a hybrid mechanism combining neighborhood masking and gamma enhancement to construct image pairs with varying illumination and noise levels for joint denoising and enhancement training. In set1, we removed the masking mechanism and trained using the original resolution images with different illumination. In set2, we applied the full preprocessing mechanism but omitted the regularization term in Equ. 14.

We implemented these settings on the LOLv1 and LOLv2-Real datasets, with the quantitative results presented in Tab. 4 and the visual comparisons shown in Fig.8.

The results indicate that removing any part of the strategy reduces performance, and the combination of both strategies is necessary to achieve optimal denoising results. In set1, the noise intensity is significantly pronounced, primarily due to the decomposition network generating identity mappings while learning the illumination map. In set2, images lose detail in underexposed regions, which is attributed to the local semantic loss caused by downsampling.

**Hybrid Piror Design.** Tab. 3 and Fig. 9 present the results of the ablation study on mixed priors. The priors are categorized into three parts: illumination prior, high-pass filtering prior, and low-pass filtering prior, which respectively capture brightness, noise, and color information. The results are worse when all priors are removed, with a notable improvement of approximately 0.6 dB when the illumination prior is included. On top of the full version, removing either the high-frequency or low-frequency components adversely affects performance, demonstrating that combining multiple informative cues achieves the best results.

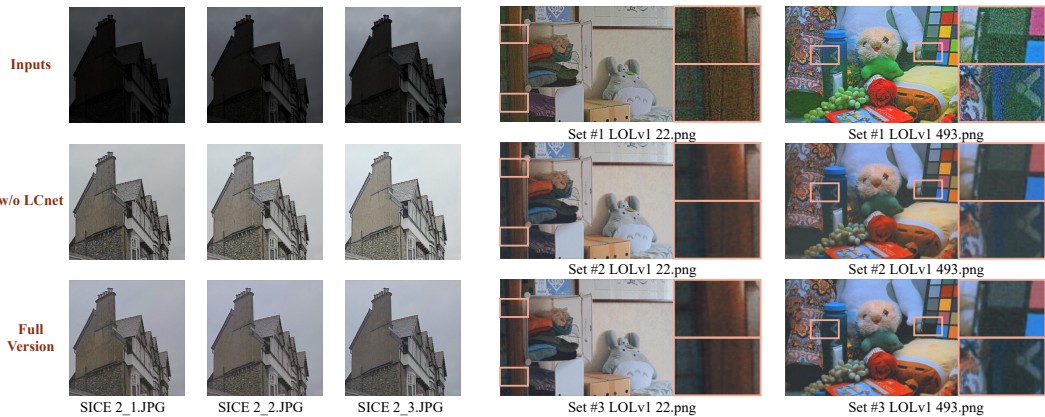

Figure 8: Left: Visualization of LCnet adaptivity experiment. Right: Visualization of denoising design ablation.

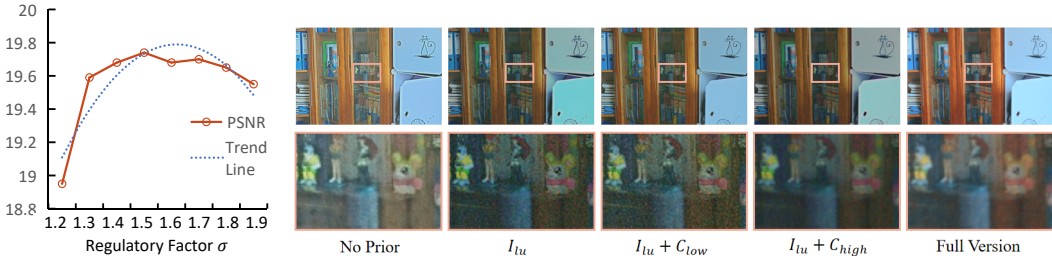

Figure 9: Left: PSNR variation with gamma enhancement factor on the LOLv1 dataset. Right: Ablation study of different physical priors.

**LCnet.** The design of LCnet aims to build an illumination-adaptive module that adjusts the illumination map to achieve the highest perceptual quality. We removed LCnet and employed a reference adjustment strategy similar to PairLIE. The visual results are shown in Fig. 8. Without the adaptive strategy, it is challenging to achieve consistent enhancement results across images with varying low-light degradations from the same scene, leading to overexposure in local regions.

**Gamma Enhancement Factor.** For the gamma enhancement operation applied to images with different illumination during pre-training, we explored which enhancement factor yields the best performance. We use $\sigma$ to regulate $\lambda$ through the formula $\lambda = \frac{1}{\sigma}$. The results are shown in Fig. 9, illustrating that the enhancement effect follows an increasing trend initially and then decreases within the $\sigma$ range of 1.2 to 1.9. At lower values, the enhanced images do not produce sufficient illumination differences with the other sub-images, which is crucial for model decomposition. At higher values, the enhancement does not conform to the assumption $R_1^{\lambda-1} = 1$ during framework inference, resulting in more complex nonlinear noise variations that negatively impact model performance. Therefore, during each iteration, we randomly sample enhancement factors within the range of $(1.3, 1.7)$ to provide the model with a broader range of feature processing options. The specific selection criteria for the control factors are detailed in the supplementary materials.

## 5 CONCLUSION

This paper tackles the challenges of low-light image enhancement and denoising, particularly in complex real-world scenarios. We propose a zero-reference framework combining self-supervised denoising via neighboring pixel downsampling and enhancement using random gamma adjustment with retinal perception theory. To address the limitations of existing methods in handling frequency-domain degradations, we introduce an RGB-space DCT-based filtering module for multi-frequency separation and a Dynamic Discrete Sequence Fusion Transformer to integrate frequency-domain priors. Experiments on real-world datasets show our method outperforms state-of-the-art techniques, offering a robust solution for low-light enhancement and denoising.

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
