# SUPPLEMENTARY MATERIALS: INTERPRETABLE UNSUPERVISED JOINT DENOISING AND ENHANCEMENT FOR REAL-WORLD LOW-LIGHT SCENARIOS

**Huaqiu Li, Xiaowan Hu, Haoqian Wang** *
Tsinghua Shenzhen International Graduate School
Tsinghua University
`lihq23@mails.tsinghua.edu.cn`

## 1 IMPLEMENTATION DETAILS

### 1.1 DATASETS

LOLv1 Wei et al. (2018) is a classic low-light dataset containing images from various scenes under different lighting conditions, comprising 500 pairs of normal-light and low-light training images and 15 pairs of testing images. LOLv2-real Yang et al. (2021) is a dataset captured in real-world scenarios by varying ISO and exposure time, containing a total of 689 training image pairs and 100 testing image pairs. Both of these datasets are real-world datasets that include noise within the images. The images in the LOL series datasets are sized 400x600 pixels, and we follow the official train-test split ratio provided in the dataset. We utilized the combined 1189 images from the v1 and v2 training sets during training on the LOL series datasets, using only the unlabeled low-light data. In the testing phase, reference metrics were calculated using the normal-light data.

SICE Cai et al. (2018) is a large-scale multi-exposure image dataset. The SICE dataset contains high-resolution multi-exposure image sequences that cover a diverse range of scenes. In the dataset, Multi-Exposure Fusion (MEF) and High Dynamic Range (HDR) techniques are used to reconstruct reference images. Through a detailed process from image capture to filtering and reference generation, 1,200 sequences were combined with 13 MEF/HDR algorithms, resulting in 15,600 fused images. After careful selection, 589 high-quality reference images and their corresponding sequences were retained for further use. We followed the same settings as those used in PairLie Fu et al. (2023) for training and dataset splitting.

SIDD Abdelhamed et al. (2018) is a real-world noisy dataset captured using smartphone cameras. For our experiments, we randomly selected low-light images from the SIDD Small dataset. The selected images were cropped into patches of size 1024x512, from which 1,500 images were used for training and 80 images for testing. Due to the dataset's inclusion of reference images captured under various camera settings, it is challenging to identify the most appropriate reference. Consequently, we utilized no-reference evaluation metrics to assess the enhancement performance.

### 1.2 TRAINING DETAILS

In the experiments, we employed the 'seed_torch' function to set the random seeds for Python, NumPy, and PyTorch to "123." The loss function is defined as $\mathcal{L} = \omega_R \mathcal{L}_R + \omega_L \mathcal{L}_L + \omega_{con} \mathcal{L}_{con} + \omega_{enh} \mathcal{L}_{enh}$, with the weights $\omega_R$, $\omega_L$, $\omega_{con}$, and $\omega_{enh}$ set to a ratio of 1:1:0.1:1. In $\mathcal{L}_{enh}$, the ratio of the components controlling exposure to those controlling color balance is 1:0.5. During the preprocessing of the training data, we randomly sampled images from the initialized dataset and cropped them into patches of size 256×256 pixels. Additionally, we applied data augmentation techniques—specifically horizontal flipping, vertical flipping, and image rotation—to mitigate overfitting. Finally, the data was normalized to the range of (0, 1) before being fed into the deep network.

## 2 DCT AND IDCT

As for frequency prior, we use channel-wise 2D DCT to convert the spatial-domain image $I$ into the frequency-domain counterpart $F$. Different spectral bands in the DCT domain encode different

---
*Corresponding author.

Table 1: The cascaded ablation study on the low-light SIDD dataset presents no-reference quantitative metrics, where the top-performing method is highlighted in **red** and the second-best method is marked in **blue**.

| Methods | Denoise-Enhance | | Enhance-Denoise | | Single Method | | |
|---|---|---|---|---|---|---|---|
| | EnlightenGAN | PairLIE | EnlightenGAN | PairLIE | EnlightenGAN | PairLIE | Ours |
| BRISQUE | 24.727 | 36.100 | 13.382 | 47.653 | 13.786 | **3.168** | **2.555** |
| CLIPIQA | 0.341 | 0.377 | 0.355 | 0.359 | **0.337** | 0.383 | **0.292** |

image visual attributes degradation representation analysis of input images:

$$F_{u,v} = \frac{2}{\sqrt{hw}} m(u)m(v) \sum_{i=0}^{h-1} \sum_{j=0}^{w-1} [I_{i,j} \cos \frac{(2i+1)u\pi}{2h} \cos \frac{(2j+1)v\pi}{2w}], m(\gamma) = \begin{cases} \frac{1}{\sqrt{2}}, \gamma=0 \\ 1, \gamma>0 \end{cases} \quad (1)$$

where the index $i$, $j$ denote the 2D coordinate in the spatial domain, while $u$, $v$ refer to the 2D coordinate in the DCT frequency domain.

By performing an inverse Discrete Cosine Transform (IDCT) on these filtered maps $F_*$, we obtain the corresponding spatial domain feature images:

$$C_{i,j}^* = \frac{2}{\sqrt{hw}} \sum_{u=0}^{h-1} \sum_{v=0}^{w-1} [m(u)m(v)F_{u,v}^* \cos \frac{(2i+1)u\pi}{2h} \cos \frac{(2j+1)v\pi}{2w}], m(\gamma) = \begin{cases} \frac{1}{\sqrt{2}}, \gamma=0 \\ 1, \gamma>0 \end{cases} \quad (2)$$

## 3 EXPERIMENTS

### 3.1 COMPARISON OF CASCADING METHODS

In the main text, we mention that most foundational frameworks for joint tasks employ multi-stage training and utilize a cascaded approach to handle each degradation task sequentially. To evaluate the performance difference between this approach and ours, we selected no-reference unsupervised denoising and enhancement methods for a cascaded implementation of this task. For unsupervised denoising, we employed the neighbor masking pipeline from Neighbor2neighbor Huang et al. (2021). Additionally, we paired two unsupervised low-light enhancement methods without explicit denoising design, EnlightenGAN Jiang et al. (2021) and PairLIE Fu et al. (2023), with the denoising method in a sequential inference. This resulted in four combinations arranged based on their processing order.

The qualitative results of this comparative experiment are shown in Tab. 1, with corresponding visual results in Fig. 1. We observed that the image quality generated by these simple cascaded combinations tends to be worse than that of single-task methods (i.e., using only enhancement techniques). Specifically, local overexposure and the generation of more complex noise patterns are common. This is due to error accumulation during the sequential processing. Low-light enhancement methods often introduce nonlinear perturbations to the original noise, making the noise pattern more challenging to model and distinguish. Additionally, after denoising, the image's statistical properties (e.g., brightness, contrast, dynamic range) may shift, further complicating the dynamic range and leading the enhancement algorithm to overcompensate or underperform.

### 3.2 GENERALIZATION EVALUATION

**Testing on unsupervised datasets.** To evaluate the generalization capability of our proposed method, we selected SCI Ma et al. (2022), PairLIE Fu et al. (2023), and RUAS Liu et al. (2021) as baseline methods. These approaches, along with our method, were applied to process datasets without ground truth(LIME Guo et al. (2016), NPE Wang et al. (2013), MEF Ma et al. (2015), DICM Lee et al. (2013) and VV Vonikakis et al. (2018)) using models trained on the LOL dataset. For a fair comparison, all methods were trained for 100 epochs. We employed two widely-used no-reference quality assessment metrics, NIQE and BRISQUE, as benchmarks to qualitatively evaluate the performance of image enhancement.

The experimental results, as shown in the Fig.2, Tab.2 and3, demonstrate that our method achieves superior performance in noise suppression, exposure stability, and natural color restoration. Compared to SCI, our approach generates fewer artifacts and noise in local details (e.g., the boxed area on the left of Fig.2). Compared to PairLIE, our method produces richer color gradations and aesthetically pleasing details, with smoother handling of light and shadows. In comparison with RUAS, our

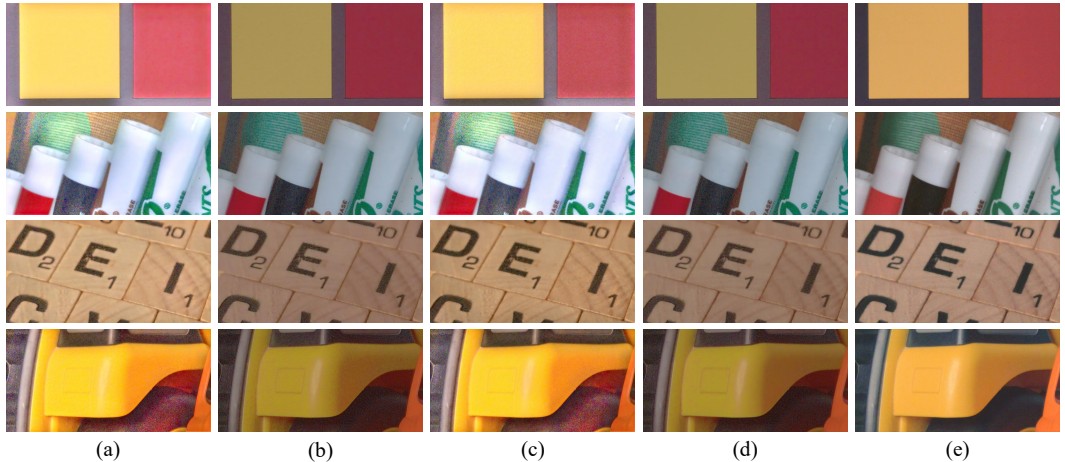

(a)     (b)     (c)     (d)     (e)

Figure 1: The visual results of the cascaded methods. (a) EnlightenGAN-Neighbor2neighbor. (b) PairLIE-Neighbor2neighbor. (c) Neighbor2neighbor-EnlightenGAN. (d) Neighbor2neighbor-PairLIE. (e)Ours.

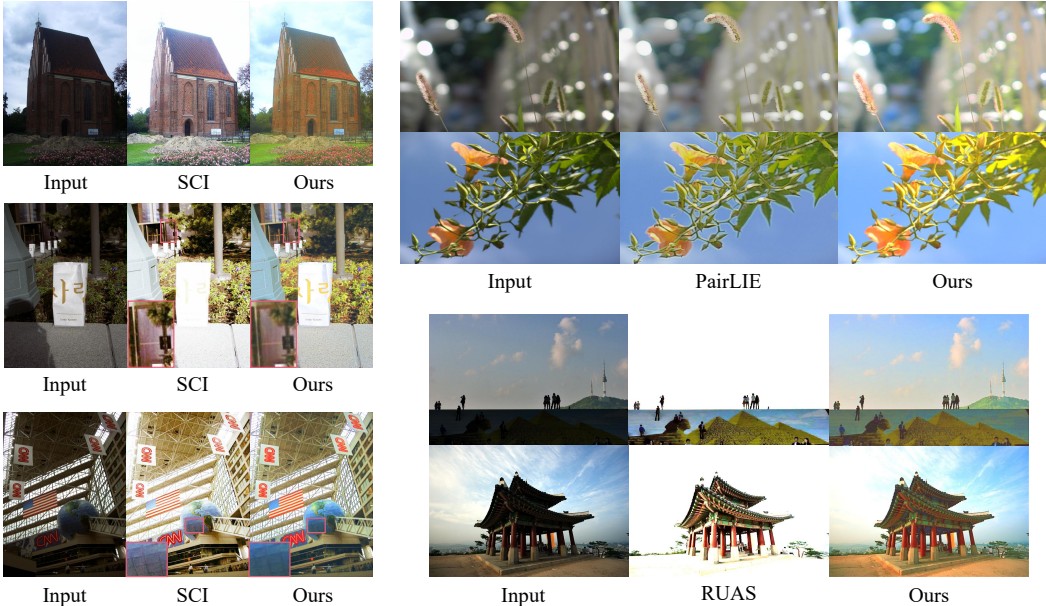

Figure 2: Qualitative results on the unsupervised dataset.

approach exhibits better generalization under the exposure conditions of the unsupervised dataset, adaptively enhancing dark regions while avoiding overexposure in originally bright areas.

**Testing on overexposed images.** In addition, we explored the generalization capability of the proposed method on overexposed images. Specifically, we employed the model trained for 100 epochs on the LOL dataset to process overexposed images. The selected test dataset consists of the overexposed samples from the test set of the Exposure-Error Dataset in Afifi et al. (2021).

As illustrated in the experimental results Fig.6, the proposed method demonstrates generalization to overexposed scenarios and achieves accurate Retinex decomposition. However, certain limitations are observed. Since our adaptive illumination adjustment module (LCNet) is trained on low-light datasets, domain discrepancies between the two tasks often result in lower output values from LCNet. This causes the enhanced image to maintain relatively high exposure levels. To address this issue, we adopt enhancement measures inspired by PairLIE, leveraging traditional techniques to adjust the decomposed illumination map. Specifically, dynamic range adjustment is employed. The final output exhibits richer color gradations and exposure levels closer to human visual perception, compared to the input image.

Table 2: Quantitative comparisons on the unsupervised dataset LIME, NPE and MEF, where the top-performing method is highlighted in **red** and the second-best method is marked in **blue**.

| Dataset | LIME | | NPE | | MEF | |
|---------|------|------|-----|------|-----|------|
| Method | NIQE↓ | BRISQUE↓ | NIQE↓ | BRISQUE↓ | NIQE↓ | BRISQUE↓ |
| RUAS | 5.376 | 28.937 | 7.060 | 49.594 | 5.423 | 33.817 |
| PairLIE | 4.569 | 23.699 | **4.137** | **21.528** | 4.288 | 28.388 |
| SCI | **4.182** | **19.701** | 4.473 | 27.657 | **3.634** | **14.399** |
| Ours | **4.109** | **16.382** | **3.802** | **17.140** | **3.758** | **18.997** |

Table 3: Quantitative comparisons on the unsupervised dataset DICM and VV, where the top-performing method is highlighted in **red** and the second-best method is marked in **blue**.

| Dataset | DICM | | VV | |
|---------|------|------|-----|------|
| Method | NIQE | BRISQUE | NIQE | BRISQUE |
| RUAS | 7.052 | 46.522 | 5,297 | 51.085 |
| PairLIE | **4.064** | 30.833 | **3.648** | 31.213 |
| SCI | 4.073 | **27.706** | **2.934** | **21.431** |
| Ours | **3.859** | **26.592** | 3.748 | **29.701** |

## 3.3 ABLATION STUDY

**The impact of the gamma coefficient control factor.** We analyzed the impact of the gamma coefficient control factor $\sigma$ on image restoration performance. Initially, as $\sigma$ increases, PSNR exhibits an upward trend, while LPIPS shows a downward trend. The optimal denoising and enhancement effects occur at $\sigma = 1.5$, where both metrics reach their respective optimal ranges. The variation in SSIM follows a similar trend, peaking at $\sigma = 1.8$. This indicates that $\sigma = 1.5$ represents the optimal point for overall image quality, with minimal distortion and superior perceptual quality, while SSIM reaches its highest value at $\sigma = 1.8$, albeit with negligible overall variation. Thus, our results suggest that $\sigma = 1.5$ is the optimal parameter for image restoration.

Furthermore, we investigated the restoration outcomes across three variable intervals. The findings indicate that when $\sigma$ is subjected to random sampling within a specified interval, the resultant restorations significantly outperform those achieved with a fixed factor at the midpoint of the interval. We propose that this random sampling approach mitigates the model's tendency to learn the identity transformation imposed by gamma nonlinear enhancement, thereby bolstering the robustness of the retinal decomposition process. Accordingly, we identified the optimal interval $(1.3, 1.7)$ for our experimental configuration.

**the impact of different masking strategies.** Additionally, we analyzed the impact of different masking strategies on the performance of the joint framework. We selected the Neighborhood Masking proposed by Neighbor2Neighbor Huang et al. (2021) and the Mean Masking introduced by ZS-N2N Mansour & Heckel (2023), as illustrated in Fig. 4. We have supplemented our work with ablation studies on masking strategies. All experiments were conducted under the same settings. We referred to the horizontal and vertical masking strategies from Noise2Fast Lequyer et al. (2022). The specific steps are as follows:

Noise2Fast-H: The image is divided into patches of size 2×1 pixels along the height direction. The top and bottom pixels within each patch are placed into the corresponding positions of two sub-images, respectively. After masking, the sub-image dimensions are $H/2 \times W$, containing 50% of the pixel information.

Table 4: The impact of the loss function on model performance evaluated on the LOLv1 and LOLv2 dataset. The data ranked first is highlighted in **red**.

| Dataset | LOLv1 | | | LOLv2 | | |
|---------|-------|------|-------|-------|------|-------|
| Setting | PSNR | SSIM | LPIPS | PSNR | SSIM | LPIPS |
| w/o $\mathcal{L}_R$ | 18.93 | 0.713 | 0.266 | 19.40 | 0.723 | 0.290 |
| w/o $\mathcal{L}_L$ | 12.61 | 0.542 | 0.725 | 12.05 | 0.492 | 0.705 |
| w/o $\mathcal{L}_{enh}$ | 7.30 | 0.130 | 0.720 | 9.12 | 0.140 | 0.699 |
| w/o $\mathcal{L}_{con}$ | 19.49 | 0.744 | 0.279 | 19.72 | 0.759 | 0.277 |
| Full Version | **19.80** | **0.750** | **0.253** | **20.22** | **0.793** | **0.266** |

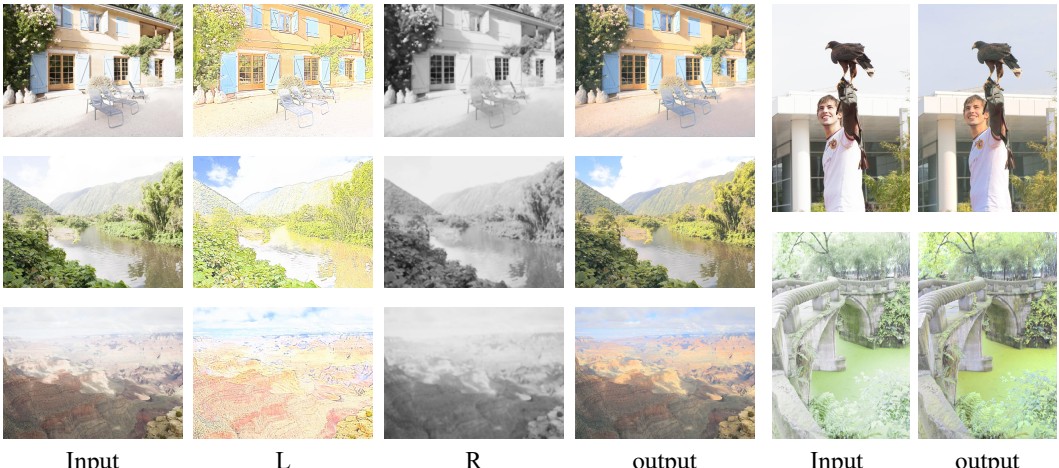

Figure 3: Visualization results on the overexposed dataset.

| $\sigma$ | PSNR | SSIM | LPIPS |
|---|---|---|---|
| 1.2 | 18.95 | 0.735 | 0.279 |
| 1.3 | 19.59 | 0.737 | 0.256 |
| 1.4 | 19.68 | 0.738 | 0.255 |
| 1.5 | **19.74** | 0.740 | **0.254** |
| 1.6 | 19.68 | 0.741 | 0.255 |
| 1.7 | 19.70 | 0.741 | 0.257 |
| 1.8 | 19.65 | **0.742** | 0.257 |
| 1.9 | 19.55 | 0.737 | 0.260 |
| 1.2-1.6 | 19.69 | 0.740 | 0.255 |
| 1.3-1.7 | **19.80** | **0.750** | **0.253** |
| 1.4-1.8 | 19.71 | 0.742 | 0.256 |

Table 5: The impact of the regulation factor $\sigma$ on model performance evaluated on the LOLv1 dataset. The data ranked first is highlighted in **red**.

| Method | Flops(G) | Time(ms) |
|---|---|---|
| Retinexnet | 14.23 | 7.37 |
| LLFormer | 3.46 | 51.99 |
| SNR-aware | 6.97 | 19.89 |
| Zero-DCE | 1.30 | **1.39** |
| RUAS | 0.05 | 3.72 |
| Clip-LIT | 4.56 | 2.10 |
| PairLIE | 5.59 | 1.70 |
| SCI | **0.01** | 1.52 |
| ours | 5.10 | 16.56 |

Table 6: Comparison of Computational Complexity and Runtime Efficiency. The data ranked first is highlighted in **red**.

Noise2Fast-W: The image is divided into patches of size 1×2 pixels along the width direction. The left and right pixels within each patch are placed into the corresponding positions of two sub-images, respectively. After masking, the sub-image dimensions are $H \times W/2$, containing 50% of the pixel information.

Mean Masking: The image is divided into patches of size 2×2 pixels. The average of the pixels along the two diagonals is computed and placed into the corresponding positions of two sub-images. After masking, the sub-image dimensions are $H/2 \times W/2$, containing 50% of the pixel information.

All three methods employ deterministic masking strategies, whereas Neighbor-masking adopts a stochastic masking approach. The experimental results are presented in Tab. 7.

The convolution operation is performed on each region.

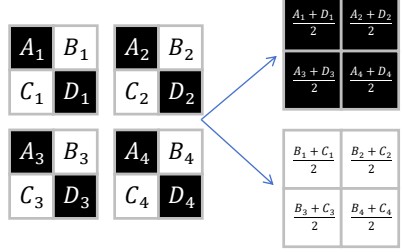

(a) Mean Masking

Randomly sampling a pixel and its neighbor.

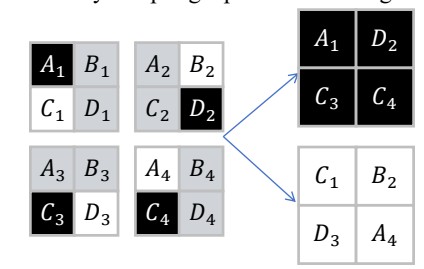

(a) Neighborhood Masking

Figure 4: The schematic diagram illustrating the mechanisms of two mask strategies.

Table 7: The ablation study on the impact of different masking strategies on the joint framework. The data ranked first is highlighted in **red**.

| Dataset | LOLv1 | | | LOLv2 | | |
|---|---|---|---|---|---|---|
| Masking Method | PSNR | SSIM | LPIPS | PSNR | SSIM | LPIPS |
| Mean Masking | 18.70 | **0.758** | 0.258 | 20.17 | 0.787 | 0.263 |
| Noise2Fast-H | 18.99 | 0.736 | 0.260 | 19.94 | 0.777 | 0.261 |
| Noise2Fast-W | 19.05 | 0.744 | 0.253 | 19.80 | 0.783 | 0.267 |
| Neighborhood Masking | **19.80** | 0.750 | **0.253** | **20.22** | **0.793** | **0.266** |

Table 8: uantitative comparisons on the real-world dataset LSRW, where the top-performing method is highlighted in **red**.

| Dataset | LSRW-Huawei | | LSRW-Nikon | |
|---|---|---|---|---|
| Method | PSNR | SSIM | PSNR | SSIM |
| RUAS | 15.74 | 0.498 | 12.21 | 0.439 |
| PairLIE | **18.99** | 0.550 | 15.52 | 0.427 |
| SCI | 15.70 | 0.428 | 14.65 | 0.407 |
| Ours | 18.94 | **0.558** | **17.61** | **0.493** |

Based on the analysis of the results above, despite the fact that the other three methods mask a smaller percentage of pixels during sampling, they fail to outperform Neighbor-masking. We attribute this outcome to the following two reasons: 1) Reduced sample diversity due to deterministic sampling: Deterministic sampling limits the variability of the training data. For instance, the other three masking strategies produce only one possible image pair per sampling, whereas Neighbor-masking generates 4×2=8 possible pairs. 2) Impact of noise correlation in real-image denoising tasks: Noise in neighboring pixels tends to exhibit correlations, which can negatively affect masked denoising. Increasing the masking ratio, thus enlarging the distance between visible pixels, helps mitigate this correlation to some extent.

Although intuitively, a higher masking ratio may result in the loss of local details and over-smoothing, the unique nature of image denoising tasks sets it apart from self-supervised image compression. Specifically, it requires additional consideration of noise correlation. After comprehensive evaluation, we opted for the current strategy. In future work, we plan to further explore alternative masking approaches.

### 3.4 COMPARISON ON LSRW DATASET

To further validate the reliability and stability of our proposed method, we conducted additional comparative experiments on the real-world LSRW dataset. The LSRW dataset consists of two subsets: Huawei smartphone and Nikon camera. We performed separate training and testing on each subset, using PSNR and SSIM as quantitative evaluation metrics. The experimental results are presented in Tab.8 and Fig.5.

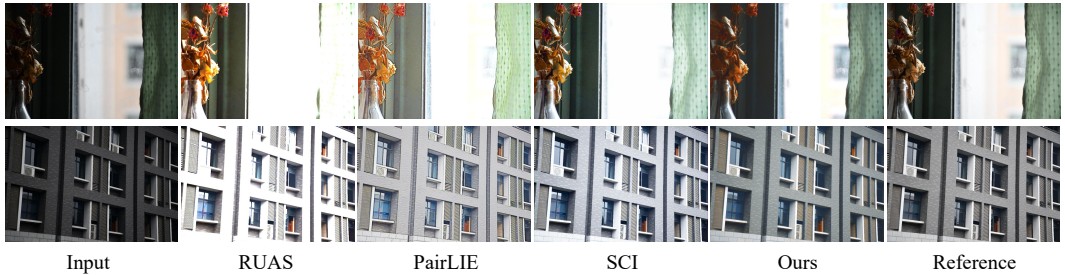

| Input | RUAS | PairLIE | SCI | Ours | Reference |

Figure 5: The visualization results on LSRW Dataset.

Compared to two other zero-reference methods, SCI and RUAS, as well as the unsupervised method PairLIE, our approach achieves superior results in both quantitative metrics and visual outcomes. From a visual perspective, RUAS and SCI suffer from the absence of adaptive illumination design, leading to overexposure or underexposure under uneven low-light conditions. In contrast, PairLIE tends to produce artifacts and excessively enhanced edges. Our method delivers the best perceptual results, addressing these challenges effectively.

## 4 THE SUPPLEMENTARY VISUAL RESULTS

Here, we provide some supplementary visual results. Fig.7 illustrates the Retinex decomposition visualization, while Fig.8, 9, 10 present the results on the LOL, SICE, and SIDD datasets, respectively.

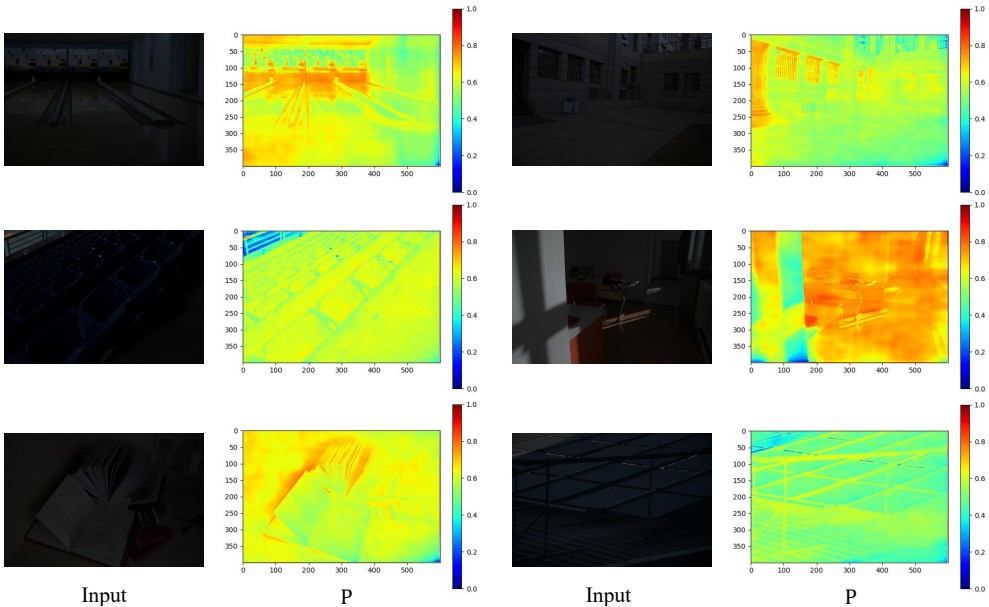

Figure 6: The visualization results of the degradation representation $P$ reveal that $P$ focuses its attention predominantly on the darker regions of the input image and effectively captures certain semantic information.

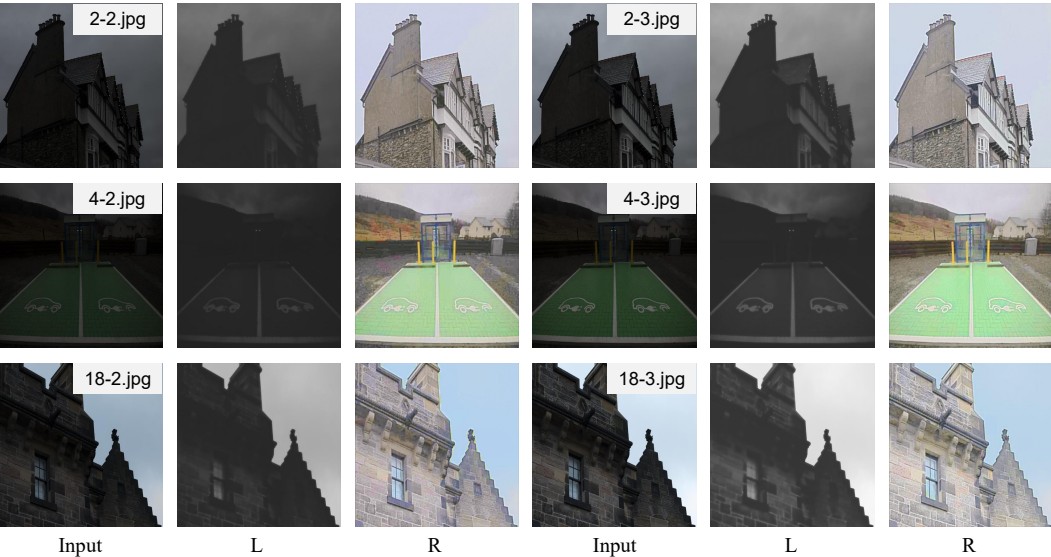

Figure 7: Retinex decomposition visual supplement results.

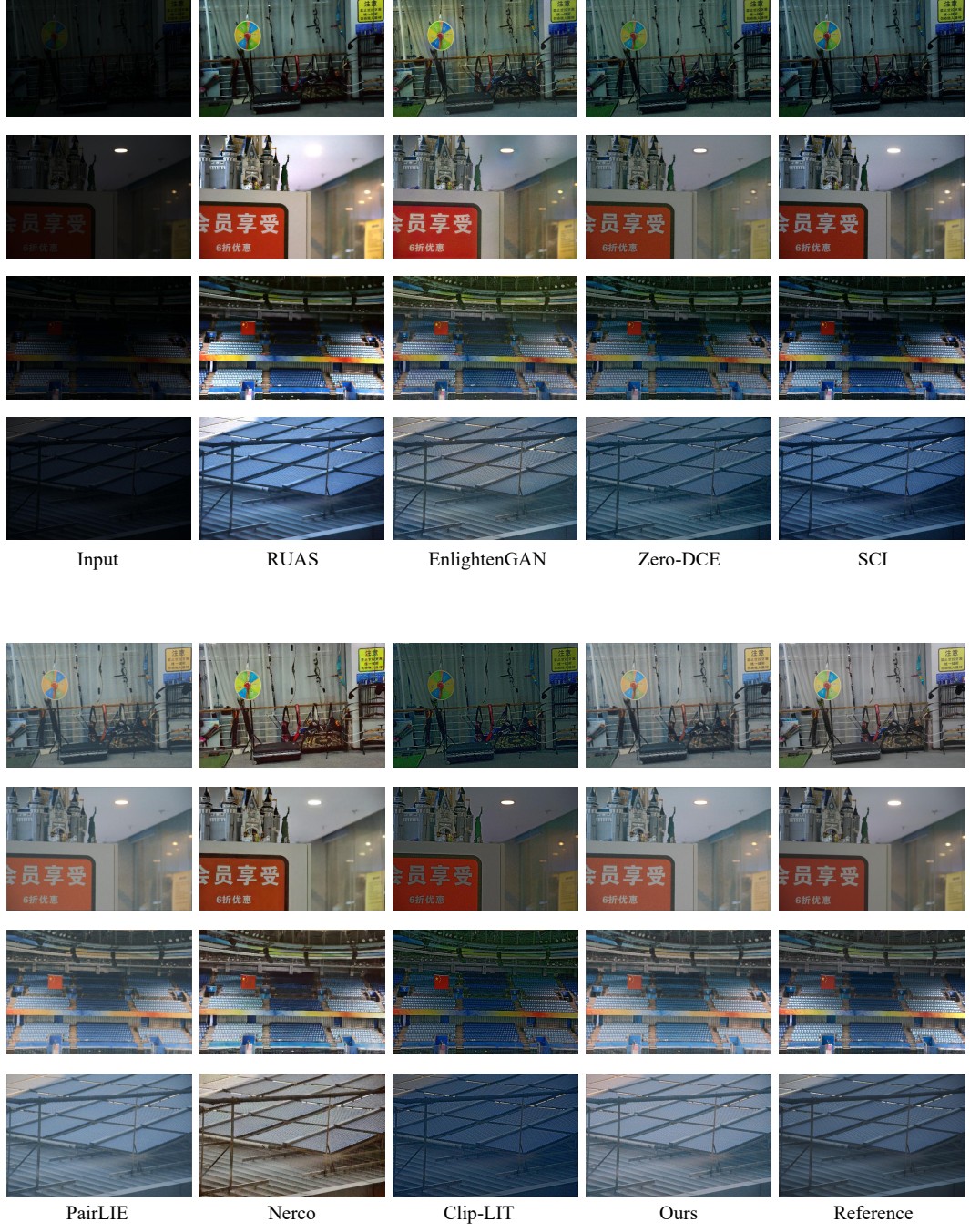

Figure 8: Additional qualitative results from comparative experiments on the LOL dataset.

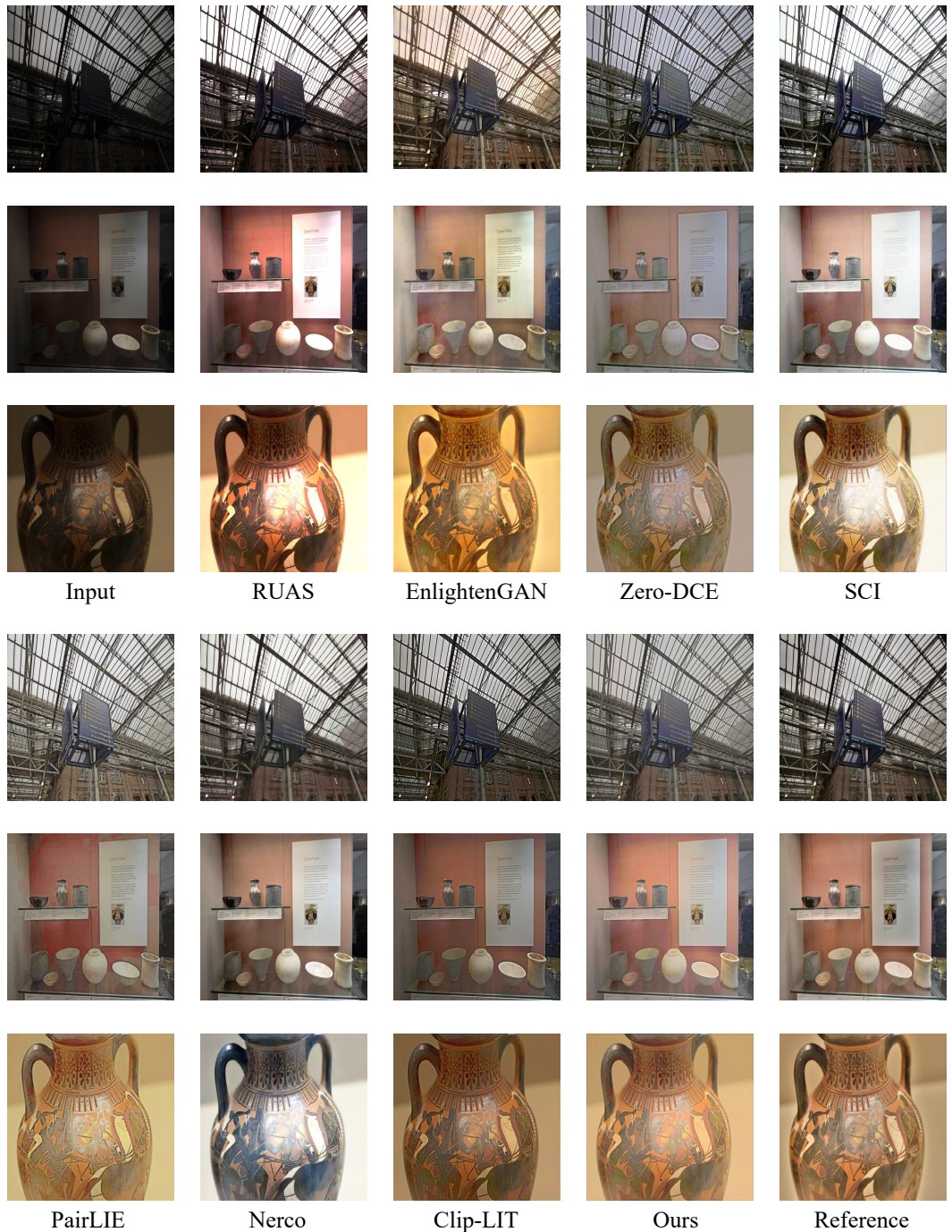

Figure 9: Additional qualitative results from comparative experiments on the SICE dataset.

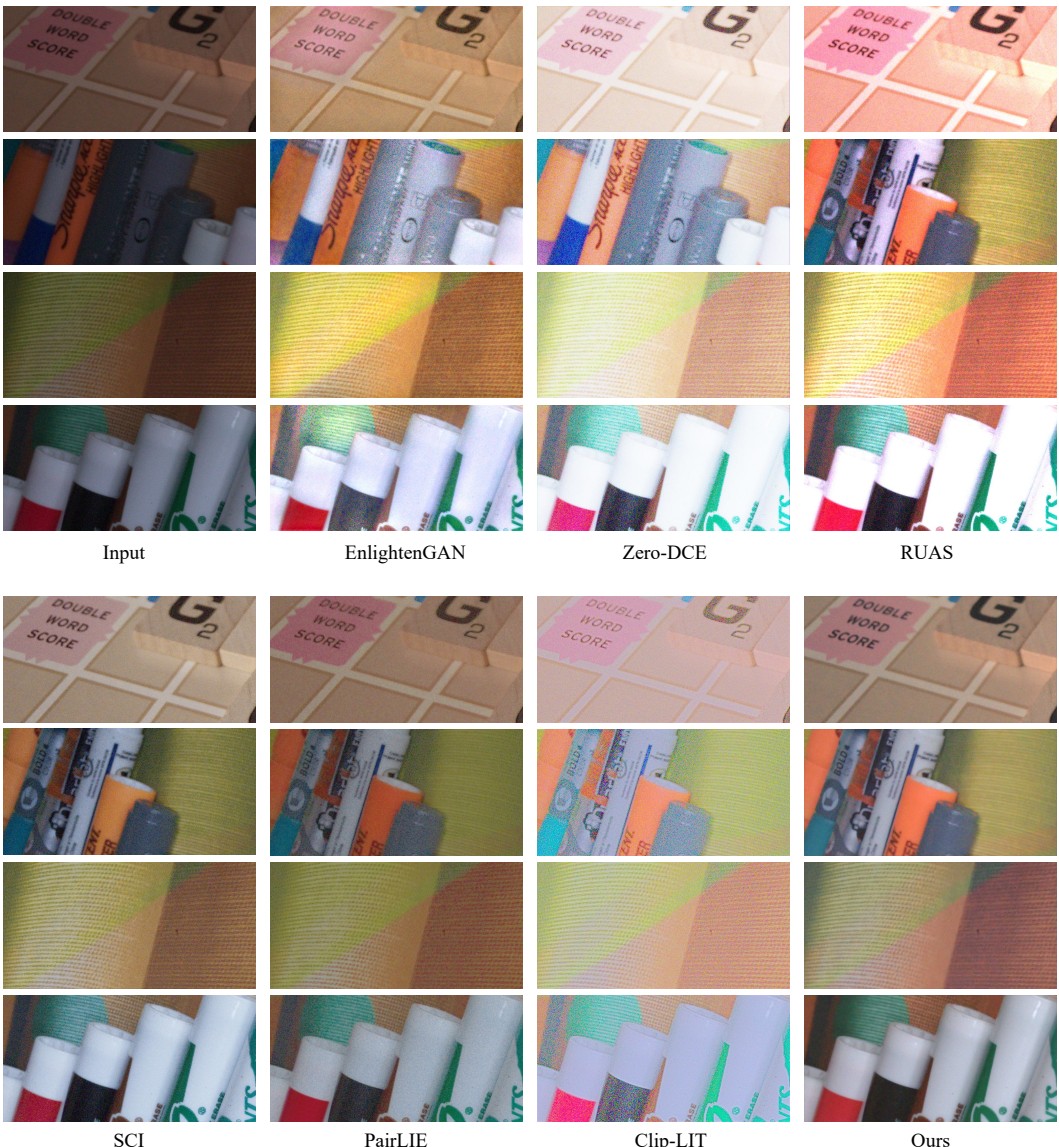

Figure 10: Additional qualitative results from comparative experiments on the sidd dataset.

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

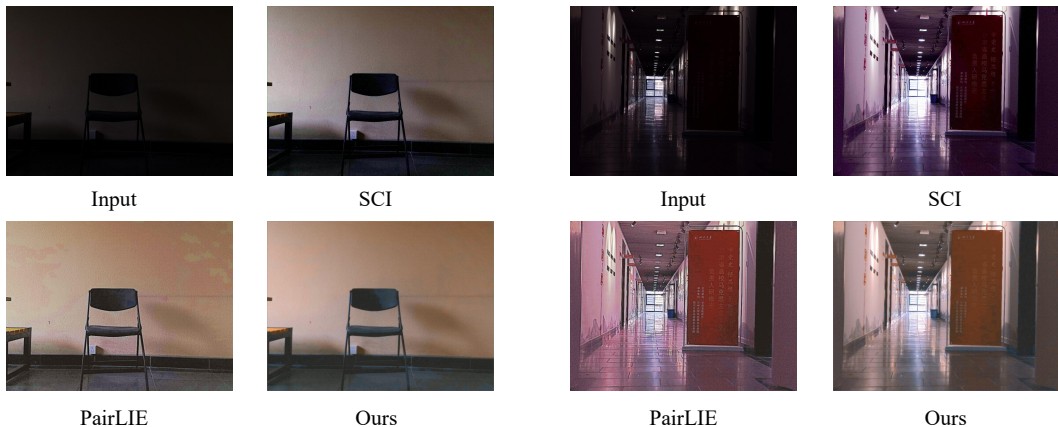

Figure 11: Additional qualitative results in edge situation with color offset and severe artifacts.

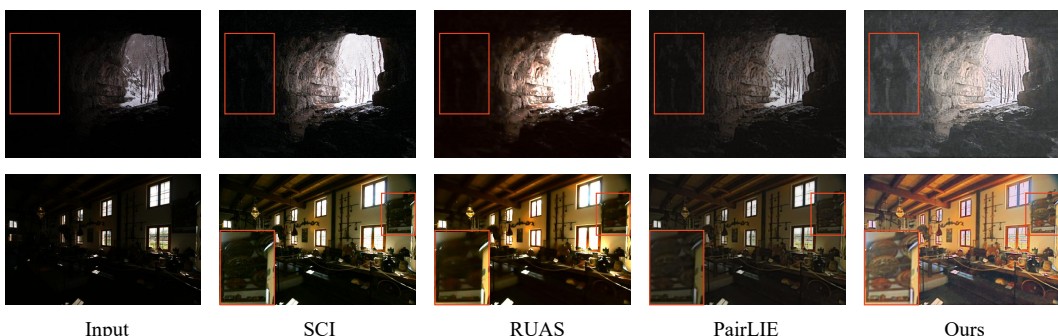

Figure 12: Additional qualitative results in edge situation of dataset MEF.

Xiaojie Guo, Yu Li, and Haibin Ling. Lime: Low-light image enhancement via illumination map estimation. *IEEE Transactions on image processing*, 26(2):982–993, 2016.

Tao Huang, Songjiang Li, Xu Jia, Huchuan Lu, and Jianzhuang Liu. Neighbor2neighbor: Self-supervised denoising from single noisy images. In *Proceedings of the IEEE/CVF conference on computer vision and pattern recognition*, pp. 14781–14790, 2021.

Yifan Jiang, Xinyu Gong, Ding Liu, Yu Cheng, Chen Fang, Xiaohui Shen, Jianchao Yang, Pan Zhou, and Zhangyang Wang. Enlightengan: Deep light enhancement without paired supervision. *IEEE transactions on image processing*, 30:2340–2349, 2021.

Chulwoo Lee, Chul Lee, and Chang-Su Kim. Contrast enhancement based on layered difference representation of 2d histograms. *IEEE transactions on image processing*, 22(12):5372–5384, 2013.

Jason Lequyer, Reuben Philip, Amit Sharma, Wen-Hsin Hsu, and Laurence Pelletier. A fast blind zero-shot denoiser. *Nature Machine Intelligence*, 4(11):953–963, 2022.

Risheng Liu, Long Ma, Jiaao Zhang, Xin Fan, and Zhongxuan Luo. Retinex-inspired unrolling with cooperative prior architecture search for low-light image enhancement. In *Proceedings of the IEEE/CVF conference on computer vision and pattern recognition*, pp. 10561–10570, 2021.

Kede Ma, Kai Zeng, and Zhou Wang. Perceptual quality assessment for multi-exposure image fusion. *IEEE Transactions on Image Processing*, 24(11):3345–3356, 2015.

Long Ma, Tengyu Ma, Risheng Liu, Xin Fan, and Zhongxuan Luo. Toward fast, flexible, and robust low-light image enhancement. In *Proceedings of the IEEE/CVF conference on computer vision and pattern recognition*, pp. 5637–5646, 2022.

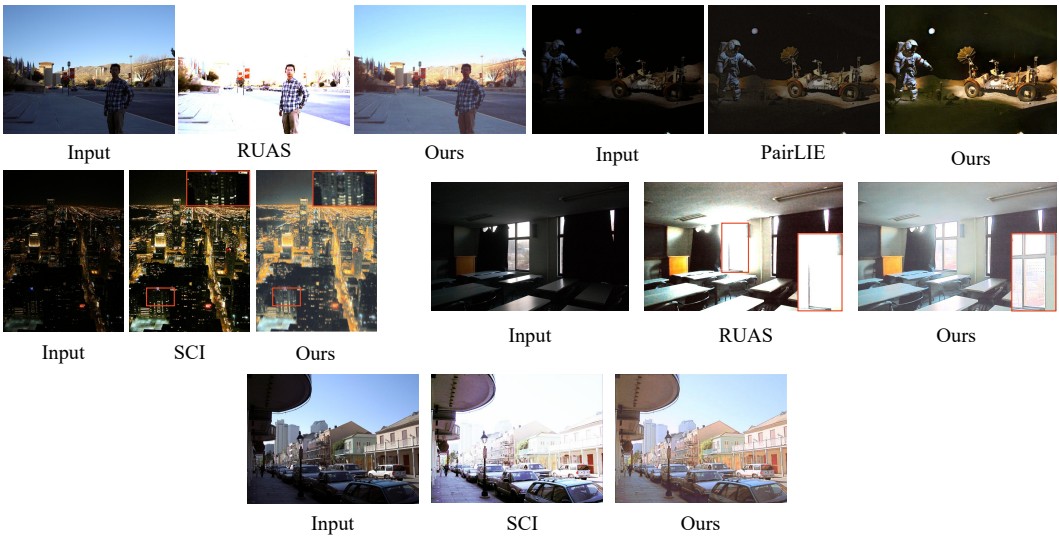

Figure 13: Additional qualitative results in edge situation of dataset DICM.

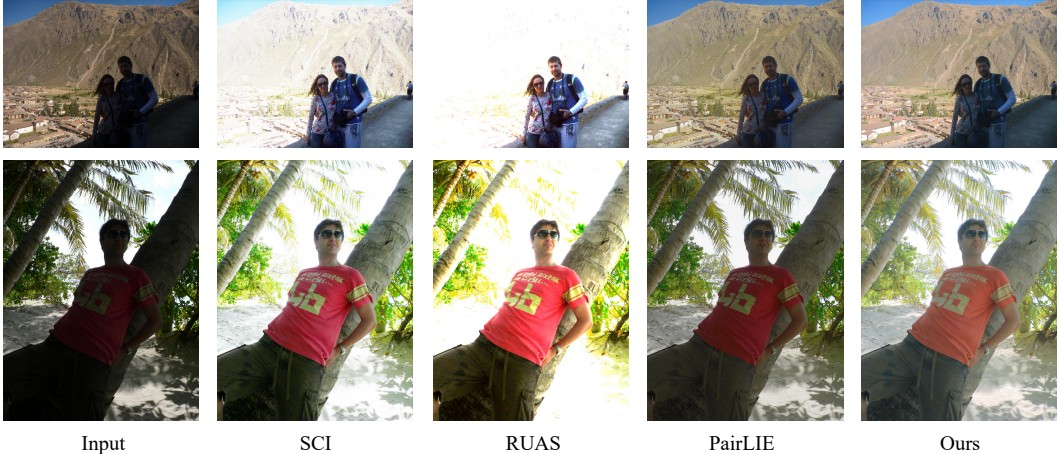

Figure 14: Additional qualitative results in edge situation of dataset VV.

Youssef Mansour and Reinhard Heckel. Zero-shot noise2noise: Efficient image denoising without any data. In *Proceedings of the IEEE/CVF Conference on Computer Vision and Pattern Recognition*, pp. 14018–14027, 2023.

Vassilios Vonikakis, Rigas Kouskouridas, and Antonios Gasteratos. On the evaluation of illumination compensation algorithms. *Multimedia Tools and Applications*, 77:9211–9231, 2018.

Shuhang Wang, Jin Zheng, Hai-Miao Hu, and Bo Li. Naturalness preserved enhancement algorithm for non-uniform illumination images. *IEEE transactions on image processing*, 22(9):3538–3548, 2013.

Chen Wei, Wenjing Wang, Wenhan Yang, and Jiaying Liu. Deep retinex decomposition for low-light enhancement. *arXiv preprint arXiv:1808.04560*, 2018.

Wenhan Yang, Wenjing Wang, Haofeng Huang, Shiqi Wang, and Jiaying Liu. Sparse gradient regularized deep retinex network for robust low-light image enhancement. *IEEE Transactions on Image Processing*, 30:2072–2086, 2021.