# OpenReview forum: "Interpretable Unsupervised Joint Denoising and Enhancement for Real-World low-light Scenarios"
_ICLR.cc/2025/Conference — ICLR 2025 Poster_

### Official Review · Reviewer_h1DE · 2024-11-02

**Soundness:** 2
**Presentation:** 3
**Contribution:** 2
**Rating:** 6
**Confidence:** 5

**Summary:**

This paper develops a zero-reference joint denoising and low-light enhancement framework designed for real-world applications.
The framework incorporates a self-supervised optimization strategy and physical priors to effectively capture complex degradations.
Grounded in Retinex theory, the optimization process generates paired low-light and enhanced images by dynamically adjusting brightness, thereby establishing mutual self-supervision.
A Discrete Cosine Transform (DCT) is employed to extract degradation representations across varying levels, facilitating degradation decomposition and removal within the frequency domain.

**Strengths:**

1. This paper presents a zero-reference joint denoising and low-light enhancement framework that operates without the need for paired or unpaired images, a crucial feature for real-world applications.

2. The framework introduces a novel neighboring pixel masking strategy and creates paired images with varying brightness levels to enable self-supervised image denoising and decomposition.

3. Experiments on real-world datasets show that the proposed framework outperforms existing zero-reference low-light image enhancement methods.

**Weaknesses:**

1. The reviewer notes that the proposed neighboring pixel masking strategy appears to draw from methods like Neighbor2Neighbor [1] and MAE [2], which are known to sometimes produce over-smoothed results. The authors are encouraged to clarify the differences between their approach and these methods, highlighting any improvements that address over-smoothing issues.

2. Conducting ablation studies on the size of the neighboring mask would be beneficial to assess its impact on preserving texture details. Such analysis could provide a clearer understanding of the mask’s effectiveness in enhancing visual quality.

3. The exposure loss function relies on a pre-defined exposure level, similar to the approach in Zero-DCE. However, in scenarios with non-uniform or overexposure, the results may tend toward overexposure and color distortion. The authors are encouraged to consider integrating an adaptive exposure adjustment strategy in future work to address these limitations.

4. While the authors claim that the proposed framework can both denoise and enhance images, the LOL dataset—characterized by slight noise degradation—may not fully validate its denoising capabilities. It would be beneficial to conduct experiments on low-light image enhancement datasets with significant noise levels, such as LSRW or other image denoising datasets, to substantiate the framework's performance in more challenging conditions.

[1] Huang T, Li S, Jia X, et al. Neighbor2neighbor: Self-supervised denoising from single noisy images[C]//Proceedings of the IEEE/CVF conference on computer vision and pattern recognition. 2021: 14781-14790.

[2] He K, Chen X, Xie S, et al. Masked autoencoders are scalable vision learners[C]//Proceedings of the IEEE/CVF conference on computer vision and pattern recognition. 2022: 16000-16009.

**Questions:**

Please refer to Weaknesses

---

> ### Author Response · Authors · 2024-11-15
>
> We thank the reviewer for the constructive feedback.
>
> >**Q1**: The authors are encouraged to clarify the differences between their approach and these methods, highlighting any improvements that address over-smoothing issues.
>
> **Answer2Q1**:Indeed, we have drawn inspiration from the masking strategies used in Neighbor2Neighbor and MAE for self-supervised training. A key distinction in our approach is the use of gamma enhancement on sub-images while constructing the mask, which allows the model to generalize to low-light noisy image enhancement. The theoretical feasibility of this method is demonstrated in Section 3.2.
>
> **Regarding the solution of over-smoothing**: we believe the over-smoothing in related works arises from not utilizing full pixel information during training. In $L_{reg}$ (Section 467 and 303), in addition to enhancing the masked images, we apply the model to the original scale images before masking. By comparing the results with previous ones, this ensures the network learns information from the original scale while avoiding identity mappings due to the loss function. The effectiveness of Lreg is supported by the ablation study in Table 4.
>
> ---
>
> >**Q3**: The authors are encouraged to consider integrating an adaptive exposure adjustment strategy
>
> **Answer2Q3**: Indeed, we have drawn inspiration from the exposure loss function from zero-DCE, which relies on manually set exposure values. However, we have made additional modifications using LC-Net to address non-uniform lighting conditions. After performing retinex decomposition on the original image, LC-Net processes the illumination map and returns a correction factor. The goal is to adaptively adjust the illumination to better align with perceptual lighting. This enables the network to not only learn the degree of enhancement but also determine the enhancement direction based on the inherent illumination, thus improving generalization across different lighting conditions. We have demonstrated this in the ablation study, as shown in the left side of Figure 8.
>
> In the future, we will consider further optimizing the exposure setting in the loss function. We appreciate your constructive feedback.
>
> ---
>
> >**Q4**: It would be beneficial to conduct experiments on low-light image enhancement datasets with significant noise levels, such as LSRW or other image denoising datasets, to substantiate the framework's performance in more challenging conditions.
>
> **Answer2Q4**: Regarding Image Denoising Validation: We have conducted experiments using the dedicated denoising dataset SIDD, with detailed results provided in Table 2 and Fig. 7 of the main manuscript. Additionally, addressing your concerns, **we have supplemented experiments on the LSRW (Huawei and Nikon) datasets. These results have been updated in the supplementary materials, specifically in Table 8 and Fig. 5 on page 6**.
>
> Thanks for the reviewer's constructive feedback. If you have any further questions or concerns, please feel free to contact us.

---

> ### Author Response · Authors · 2024-11-17
>
> > Q2: provide a clearer understanding of the mask’s effectiveness in enhancing visual quality
>
> To better address your concerns, we have supplemented our work with ablation studies on masking strategies. All experiments were conducted under the same settings. We referred to the horizontal and vertical masking strategies from Noise2Fast[1] and the mean masking strategy from ZS-N2N[2]. The specific steps are as follows:
>
> **Noise2Fast-H:** The image is divided into patches of size 2×1 pixels along the height direction. The top and bottom pixels within each patch are placed into the corresponding positions of two sub-images, respectively. After masking, the sub-image dimensions are \(H/2 x W\), containing 50% of the pixel information.
>
> **Noise2Fast-W:** The image is divided into patches of size 1×2 pixels along the width direction. The left and right pixels within each patch are placed into the corresponding positions of two sub-images, respectively. After masking, the sub-image dimensions are \(H x W/2\), containing 50% of the pixel information.
>
> **Mean Masking:** The image is divided into patches of size 2×2 pixels. The average of the pixels along the two diagonals is computed and placed into the corresponding positions of two sub-images. After masking, the sub-image dimensions are \(H/2 x W/2\), containing 50% of the pixel information.
>
> All three methods employ deterministic masking strategies, whereas **Neighbor-masking** adopts a stochastic masking approach.
>
> | Method           | Mask Percentage | LOLv1-PSNR(dB) | LOLv1-SSIM | LOLv1-LPIPS | LOLv2-PSNR(dB) | LOLv2-SSIM | LOLv2-LPIPS |
> | ---------------- | --------------- | -------------- | ---------- | ----------- | -------------- | ---------- | ----------- |
> | Mean-Masking     | 50%             | 18.7           | 0.758      | 0.258       | 20.17          | 0.787      | 0.263       |
> | Noise2Fast-H     | 50%             | 18.99          | 0.736      | 0.260       | 19.94          | 0.777      | 0.261       |
> | Noise2Fast-W     | 50%             | 19.05          | 0.744      | 0.253       | 19.80          | 0.783      | 0.267       |
> | Neighbor-Masking | 75%             | 19.80          | 0.750      | 0.253       | 20.22          | 0.793      | 0.266       |
>
> Based on the analysis of the results above, despite the fact that the other three methods mask a smaller percentage of pixels during sampling, they fail to outperform Neighbor-masking. We attribute this outcome to the following two reasons:
>
> 1. **Reduced sample diversity due to deterministic sampling:** Deterministic sampling limits the variability of the training data. For instance, the other three masking strategies produce only one possible image pair per sampling, whereas Neighbor-masking generates 4×2=8 possible pairs.
>
> 2. **Impact of noise correlation in real-image denoising tasks:** Noise in neighboring pixels tends to exhibit correlations, which can negatively affect masked denoising. Increasing the masking ratio, thus enlarging the distance between visible pixels, helps mitigate this correlation to some extent.
>
> Although intuitively, a higher masking ratio may result in the loss of local details and over-smoothing, the unique nature of image denoising tasks sets it apart from self-supervised image compression. Specifically, it requires additional consideration of noise correlation. After comprehensive evaluation, we opted for the current strategy. In future work, we plan to further explore alternative masking approaches. We appreciate your constructive feedback and will include these latest experiments in the revised version.
>
> [1] Lequyer, Jason, et al. "A fast blind zero-shot denoiser." Nature Machine Intelligence 4.11 (2022): 953-963.
>
> [2] Mansour, Youssef, and Reinhard Heckel. "Zero-shot noise2noise: Efficient image denoising without any data." Proceedings of the IEEE/CVF Conference on Computer Vision and Pattern Recognition. 2023.

---

> > ### Comment · Reviewer_h1DE · 2024-11-22
> >
> > Thank you for the authors' response. It has mostly addressed my concerns, and I have accordingly raised my rating to 6.

---

> > > ### Author Response · Authors · 2024-11-22
> > >
> > > Thank you for your recognition and the score increase. We sincerely appreciate your support for our work. Should you have any further questions, we are always ready for an active discussion.

---

### Official Review · Reviewer_wCSY · 2024-11-03

**Soundness:** 3
**Presentation:** 2
**Contribution:** 3
**Rating:** 8
**Confidence:** 5

**Summary:**

This manuscript introduces an interpretable, unsupervised framework designed to enhance and denoise real-world low-light images without the need for paired training data. The proposed method leverages physical imaging principles and retinex theory to derive a training strategy based on paired sub-images with varying illumination and noise levels. The framework employs the Discrete Cosine Transform (DCT) to perform frequency domain decomposition in the sRGB space.

**Strengths:**

1. It is technically sound to incorporate the neighbor2neighbor denoiser with the task of unsupervised LLIE
2. The authors provided code in the supplementary materials.

**Weaknesses:**

1. In Equation (8), the term P1 is introduced without prior explanation, and it appears that the FIcoder is omitted in the process. Additionally, while the authors describe P1 in Equation (15), the description is still quite vague. Could the authors please clarify the shape of P1, how it is learned, what constraints are used, and whether it can be visualized?

2. If Equations (9) and (13) are identical to the standard DCT/IDCT transformations, there is no need to present them in the paper.

3. In Equation (14), the L_{reg} term is not explained. Could the authors please provide an explanation for this term?

4. In Equation (15), what is the difference between the first and third terms? Additionally, the authors are requested to show several decomposed illumination and reflection maps to demonstrate that the constraints in this equation are sufficient to learn reasonable Retinex decomposition results.

5. On Line 150, the reference year for noise2noise is incorrect; the paper was published in 2018, not 1803.

6. Figures 1, 2, 3, and 5 are not explained in the main text (at least I could not find them), while Figure 4 is explained but not correctly referenced.

7. In Figure 6, I do not see the advantage of the author's method over the SCI method, and the authors avoid explaining this point in the paper.

8. In Figure 7, the author's method still leaves noticeable noise, which I did not see in the Neighbor2Neighbor comparative experiment. Is there an issue with the author's training process?

9. The authors should introduce qualitative comparisons on an independent test set to demonstrate the superiority of their generalization performance. I personally believe this is a key experiment to distinguish whether a low-light enhancement model is valuable, and it should also be an advantage of unsupervised methods. Recommended datasets include LIME, NPE, MEF, DICM, and VV, as used in the Retinexformer.

**Questions:**

See the weakness.

---

> ### Author Response · Authors · 2024-11-15
>
> Thank you for your constructive feedback. Below are our responses, which we hope will address your concerns.
>
> **Q1**: Clarify the shape of P1,how it is learned, what constraints are used, and whether it can be visualized?
>
> **Answer2Q1**: The specific calculation process of $P_1$ can be found in Section 3.3, "Frequency-Illumination Prior Encoder."$P_1$ is obtained by concatenating the spectral maps from the DCT decomposition along the channel dimension, followed by deep convolutional learning. It is jointly trained end-to-end with the main network. In our experiments, $P_1$ is set to have 64 channels, with the same size of $H,W$ as the input image, which can be visualized. We plan to present these results in the supplementary materials.
>
> **Q3**: Could the authors please provide an explanation for L_{reg}?
>
> **Answer2Q3**: Due to space limitations, the explanation of $L_{reg}$ has been placed in Section 4.3, "Ablation Study," specifically at line 467 in the main text.
>
> **Q4**: what is the difference between the first and third terms of equ.15?
>
> **Answer2Q4**: In the third item, we applied gradient freezing during the calculation of $L_1$ using the function $\text{L1.detach()}$, which helps stabilize training. We will clarify this point in the revised version. A sample decomposition result is shown in Figure 2, and additional visual results will be included in the supplementary materials.
>
> **Q7**: What's the author's method over the SCI method in Fig.6?
>
> **Answer2Q7**: In Figure 6, our results are closer to the reference image in terms of color, while SCI's results exhibit lower color saturation. More visual comparisons between our method and SCI can be found in Figures 3 and 4 of the supplementary materials.
>
> **Q8**: about the Neighbor2Neighbor comparative experiment
>
> **Answer2Q8**: We would like to clarify that our experiments were conducted on the SIDD-srgb dataset, while the results presented in the official Neighbor2Neighbor paper were obtained using the SIDD-raw dataset. Images in the SRGB domain exhibit more complex noise characteristics due to the camera processing pipeline, and thus we aim to compare the generalization ability of algorithms under more challenging conditions. In Tables 1 and Figures 1 of the supplementary materials, we compare our method with the denoising-enhancement method based on Neighbor2Neighbor. The results demonstrate a clear advantage of our approach. We have also successfully reproduced Neighbor2Neighbor, ensuring the correctness of our training process.
>
> **Q9**: introduce qualitative comparisons to demonstrate generalization performance
>
> **Answer2Q9**: We have added experiments on the LIME, NPE, MEF, DICM, and VV datasets, with the comparison methods selected being trained on the LOL dataset. In the revised version of the paper, we will include more visual comparison results (presented similarly to those in RetinexFormer).
>
> | **Method** |   | **LIME** |   | **NPE** |  | **MEF** | | **DICM** |  | **VV**  |
> |------------|-------|----------|--------|---------|-------|---------|-------|----------|-------|---------|
> |            | NIQE$\downarrow$  | BRSIQUE$\downarrow$  | NIQE$\downarrow$   | BRSIQUE$\downarrow$ | NIQE$\downarrow$  | BRSIQUE$\downarrow$ | NIQE$\downarrow$  | BRSIQUE$\downarrow$  | NIQE$\downarrow$  | BRSIQUE$\downarrow$ |
> | RUAS       | 5.376 | 28.937   | 7.060  | 49.594  | 5.423 | 33.817  | 7.052 | 46.522   | 5,297 | 51.085  |
> | PairLIE    | 4.569 | 23.699   | 4.137  | 21.528  | 4.288 | 28.388  | 4.064 | 30.833   | 3.648 | 31.213  |
> | SCI        | 4.182 | 19.701   | 4.473  | 27.657  | **3.634** | **14.399**  | 4.073 | 27.706   | **2.934** | **21.431**  |
> | Ours       | **4.109** | **16.382**   | **3.802**  | **17.140**  | 3.758 | 18.997  | **3.859** | **26.592**   | 3.748 | 29.701  |
>
> **Answer2Q2&5&6**: Thank you for your valuable writing suggestions. Due to time constraints during the submission of the paper, several details were not fully refined. We plan to submit our revised version next week, where we will address the improvements. We appreciate your constructive feedback.

---

> > ### Comment · Reviewer_wCSY · 2024-11-17
> >
> > 1. **General Suggestion**: It appears that modifications to the main paper and supplementary materials are allowed during the rebuttal period. In light of this, could the authors please include the promised visualizations in the supplementary materials during this time? These visualizations are crucial for a comprehensive assessment and may significantly influence my evaluation.
> >
> >
> > 2. **Comment on Answer2Q1**: While the definition of $P$ is provided at the end of Section 3.3, the authors are still suggested to introduce the definitions of all variables prior to their first usage. Additionally, the relationship between $P_1$ and $P$ should be explicitly clarified.
> >
> >
> > 3. **Comment on Answer2Q3**: Placing the definition of a loss function in the ablation study section is not suggested. The explanation of $L_{\text{reg}}$ remains unclear and requires further elaboration.
> >
> >
> > 4. **Comment on Answer2Q7**: I have carefully examined the comparison between the results of the proposed method and the SCI method as presented in Figure 6 of the main paper and Figures 3 and 4 of the supplementary materials. The observed improvements over SCI are marginal and unstable. I remain skeptical about the superiority of the proposed method.
> >
> >
> > 5. **Comment on Answer2Q8**: Regarding Figure 1 in the supplementary materials, which results correspond to the authors’ method? Assuming they are represented in subfigure (e), the outputs still exhibit noticeable color distortions, such as that observed in the first row of Figure 1.

---

> ### Author Response · Authors · 2024-11-20
> **reply to reviewer wCSY**
>
> We sincerely appreciate your time and constructive feedback. Over the past few days, we have made corresponding revisions to both the main manuscript and the supplementary materials, and we have submitted the updated version. We hope these changes address your concerns effectively.
>
> **Response to General Suggestions**:
>
> We have organized the relevant visual results and included them in the supplementary materials. Specifically, **the visualization of $P$ is presented in Fig. 6 on page 7, while Fig. 7 provides additional retinal decomposition results**. Using low-light images from the same scene, our method successfully separates illumination maps of varying brightness levels and consistent reflection maps.
>
> **The visualization results for LIME, NPE, MEF, DICM, and VV have been added to Fig. 2 on page 3**. Compared with SCI, our approach demonstrates better handling of lighting details and noise suppression. When compared with PairLIE, our method preserves vibrant colors and appropriate lighting effects. Additionally, compared with RUAS, our approach processes overexposed regions, such as skies, more effectively.
>
> Beyond the visual results you previously mentioned, we have also included experiments on overexposed datasets (sup. Fig. 3) and comparative experiments with LSRW (sup. Fig. 5) to further demonstrate the reliability and generalization capability of our method.
>
> ---
>
> **Answer to Comment 3**:
>
> The results on the unsupervised datasets LIME, NPE, MEF, DICM, and VV demonstrate that **our method generalizes better to scenes with uneven illumination, achieving superior visual outcomes compared to SCI**. This improvement stems from the fact that methods like SCI, while highly concise, lack designs for illumination adaptability. Consequently, **they are prone to overexposure or underexposure when handling images with illumination conditions differing from the training data**. To address this issue, we proposed LCnet, as detailed in the discussion with Reviewer h1DE in **Answer to Q3**.  The previous visualization results were selected randomly. We will include more images that better showcase the advantages of our method in the final version. Thank you for this valuable suggestion.
>
> ---
>
> **Answer to Comment 4**:After a thorough review and comparison, we found that the color shift was caused by a participant using screenshots of the display while painting this figure. This may have introduced some color discrepancies because of display device. We have redrawn the relevant results using the original images. We sincerely appreciate your feedback.
>
> ---
>
> **Answer to Comments 1 & 2**: We have added explanations for $P, P1$, and $P2$ in Line 280 on Page 6 of the main text. Additionally, an explanation for *Lreg* has been added in Line 310. Other issues, including references to previously mentioned figures, citations, and related errors in the main text, have been corrected one by one.
>
> Thanks for the reviewer's constructive feedback. If you have any further questions or concerns, please feel free to contact us.

---

> > ### Comment · Reviewer_wCSY · 2024-11-24
> >
> > 1. The revised version of the paper is notably more comprehensible and easier to follow. However, considering the constraints of the current review period, I recommend that the authors further refine the formulation in the manuscript before submitting the camera-ready version. For instance, instead of the phrase: *"..., we apply gamma correction to $D_2(I)$ and get $\bar{D}_2(I)$"*, a more concise alternative could be: *"..., we apply gamma correction to $D_2(I)$, resulting in $\bar{D}_2(I) = {D_2(I)}^\gamma$."* Additionally, it is advisable to avoid overly lengthy inline equations, such as the one on Line 215, and to use Greek or Calligraphic letters for denoting subnetworks (e.g., FIcoder, REFnet, LUMnet, LCnet, etc.). Furthermore, in the formulas, variables should be presented in italics, while functions and logical symbols should be written in an upright (roman) font. For example, terms like $low$, $high$, $if$, $else$, and $mean$ should all be formatted in upright (roman) font for clarity and adherence to mathematical conventions.
> >
> > 2. The qualitative results provided for the unsupervised dataset include examples that do not appear to be sufficiently challenging. To better evaluate the robustness of the proposed method, I suggest including results for more difficult cases, such as 10.JPG, 12.JPG, 15.JPG, 20.JPG, and 29.JPG from the DICM dataset, Cave.png and Farmhouse.png from the MEF dataset, and P1010234.png and P1090815.png from the VV dataset. Given the improvements made to the manuscript following the revision, I have decided to raise the score. However, the final score will depend on the authors’ ability to address these challenging cases and demonstrate the robustness of their method.

---

> ### Author Response · Authors · 2024-11-25
>
> We sincerely appreciate your valuable writing suggestions. We will further refine and integrate these improvements in future versions.
>
> Regarding the test results for challenging scenarios, additional findings have been included in the supplementary materials (see Figs. 12, 13, and 14). From an overall visual perspective, our method outperforms existing unsupervised approaches. Meanwhile, we have identified certain strengths and limitations.
>
> As shown in Fig. 2, both Cave.png and Farmhouse.png exhibit "dead black" regions where pixel values are very close to zero, making them difficult to adjust and enhance. Our method effectively restores details in these "dead black" areas, which is particularly beneficial for downstream tasks such as object detection. However, the contrast of the images may be slightly reduced, impacting overall aesthetic appeal. Moving forward, we plan to balance the trade-off between visual aesthetics and enhancement quality to further improve our method.
>
> Thank you for your feedback. Please feel free to reach out if you have any additional concerns or suggestions.

---

> ### Comment · Reviewer_wCSY · 2024-11-25
>
> Considering that the proposed method demonstrates decent generalization capability in challenging real-world scenarios, I have raised my final score to 8. However, the authors should carefully revise their manuscript to enhance its clarity and readability. The organization of figures and experiments in the main text and supplementary materials should also be restructured based on their importance. Additionally, I encourage the authors to maintain an open-source repository for their method and develop an interactive demo to contribute to the field of unsupervised low-light image enhancement.

---

> ### Author Response · Authors · 2024-11-25
>
> Thank you very much for your recognition and support. We will carefully revise the paper based on your feedback and further refine and release our work.
>
> If you have any additional suggestions or questions, please do not hesitate to contact us.
>
> Best regards,
>
> Authors of submission 1007

---

### Official Review · Reviewer_v7Se · 2024-11-03

**Soundness:** 3
**Presentation:** 2
**Contribution:** 2
**Rating:** 6
**Confidence:** 4

**Summary:**

In this article, to address the complex degradation issues of images in low-light scenarios, the author proposes a training strategy based on physical imaging principles and retinal theory. Specifically, first, the author derives a novel training strategy from paired images under varying lighting conditions and noise levels; second, the author introduces an implicit guidance hybrid representation strategy through DCT transformation, effectively separating complex composite degradation; finally, the author develops a retinal decomposition network based on the implicit degradation representation mechanism. The article demonstrates the effectiveness and reliability of the proposed method through extensive experiments.

**Strengths:**

1. The author proposes a novel retinal decomposition network based on an implicit degradation representation mechanism in the article, which shows certain effectiveness in image enhancement for low-light scenarios;
2. The language of the article adheres to English writing standards and is fluent;
3. The article provides mathematical proofs for each module of the network, making the logic of the article strong;

**Weaknesses:**

1. On page 4, lines 175-180, the author proposes the use of Discrete Cosine Transform (DCT) to separate different frequencies of images (including chromatic, semantic information, edge contours, and noise intensity). What is the theoretical basis for this approach? Or is it more of an empirical practice? If so, would the performance of the model be affected if different methods were used to decompose the feature maps after Discrete Cosine Transform?
2. On page 6, lines 322-333, the author presents the loss function of the model. However, during the experiments, the author seems to have not conducted an in-depth exploration between the loss function and the performance of the model. If such research were included, it would make the argument of the paper more complete.

**Questions:**

1. On page 4, lines 175-180, the author proposes the use of Discrete Cosine Transform (DCT) to separate different frequencies of images (including chromatic, semantic information, edge contours, and noise intensity). What is the theoretical basis for this approach? Or is it more of an empirical practice? If so, would the performance of the model be affected if different methods were used to decompose the feature maps after Discrete Cosine Transform?
2. On page 6, lines 322-333, the author presents the loss function of the model. However, during the experiments, the author seems to have not conducted an in-depth exploration between the loss function and the performance of the model. If such research were included, it would make the argument of the paper more complete.

---

> ### Author Response · Authors · 2024-11-17
>
> Thank you for your constructive feedback and recognition. Below are our responses, which we hope will address your concerns.
>
> **Q1**: the author proposes the use of Discrete Cosine Transform (DCT) to separate different frequencies of images (including chromatic, semantic information, edge contours, and noise intensity). What is the theoretical basis for this approach? Or is it more of an empirical practice?
>
> **Answer2Q1**: The Discrete Cosine Transform (DCT) has been extensively analyzed and validated in the context of traditional algorithms (refer to [1][2][3]), encompassing a variety of applications such as noise and contour texture separation, chroma modulation and enhancement, and luminance-chroma decomposition, among others. Meanwhile, we also discussed with **Reviewer SCTr** the **differences and connections between our method and the latest frequency-domain decomposition approaches**. For more details, please refer to **Answer2W2**.
>
> ---
>
> [1] Buemi, Antonio, et al. "Chroma noise reduction in DCT domain using soft-thresholding." EURASIP Journal on Image and Video Processing 2010 (2011): 1-13.
>
> [2] Bilcu, Radu Ciprian, Sakari Alenius, and Markku Vehvilainen. "Combined de-noising and sharpening of color images in DCT domain." 2009 16th International Conference on Digital Signal Processing. IEEE, 2009.
>
> [3] Yu, Guoshen, and Guillermo Sapiro. "DCT image denoising: a simple and effective image denoising algorithm." Image Processing On Line 1 (2011): 292-296.

---

> ### Author Response · Authors · 2024-11-17
>
> **Q2**: the author seems to have not conducted an in-depth exploration between the loss function and the performance of the model. If such research were included, it would make the argument of the paper more complete.
>
> **Answer2Q2**: In the initial submission of our paper, we prioritized conciseness and omitted detailed discussions on the loss function design, as it was not the primary focus of our contributions. However, in response to your concerns, we have conducted additional ablation studies on the loss function, the results of which are now included and will be incorporated into the revised version of the paper. All experiments related to the loss function were uniformly conducted using the LOL dataset for training and evaluated on the official test set of LOLv1.
>
> | Version             | LOLv1-PSNR (dB)      | LOLv1-SSIM       | LOLv1-LPIPS      | LOLv2-PSNR (dB) | LOLv2-SSIM       | LOLv2-LPIPS      |
> |---------------------|---------------------|------------------|------------------|------------------|------------------|------------------|
> | w/o $L_R$          | 18.93              | 0.713            | 0.266            | 19.40           | 0.723            | 0.290            |
> | w/o $L_L$          | 12.61              | 0.542            | 0.725            | 12.05           | 0.492            | 0.705            |
> | w/o $L_{enh}$      | 7.30               | 0.130            | 0.720            | 9.12            | 0.140            | 0.699            |
> | w/o $L_{con}$      | 19.49              | 0.744            | 0.279            | 19.72           | 0.759            | 0.277            |
> | full version       | **19.80**          | **0.750**        | **0.253**        | **20.22**       | **0.793**        | **0.266**        |
>
>
> The experimental results show that $L_L$ and $L_{enh}$ are the fundamental loss terms, closely related to whether the algorithm works effectively, while $L_R$ and $L_{con}$ are enhancement loss terms, which provide further improvements to the algorithm's performance.
>
> Thanks for the reviewer's constructive feedback and recognition. If you have any further questions or concerns, please feel free to contact us.

---

### Official Review · Reviewer_SCTr · 2024-11-03

**Soundness:** 3
**Presentation:** 2
**Contribution:** 2
**Rating:** 6
**Confidence:** 3

**Summary:**

This paper proposes an interpretable unsupervised joint denoising and enhancement method suitable for real-world low-light scenes. Based on the physical imaging principles and Retinex theory, discrete cosine transform (DCT) is used for frequency domain decomposition, and an implicitly guided hybrid representation strategy is introduced to effectively separate complex compound degradations, thereby realizing the detection and optimization of complex degradation problems caused by low-light conditions. The authors conduct experiments to verify the effectiveness of the proposed method.

**Strengths:**

1. The authors explore the difficulties traditional methods face in dealing with complex degradation issues such as noise, brightness, and contrast, and propose solutions that significantly outperform existing technologies.
2. Technically, the authors introduce a spatial frequency domain filtering module that uses discrete cosine transformfor explicit multi-band separation, which facilitates the decomposition of enhanced images into illumination and reflectance maps.
3. Experiments depict the effectiveness of the proposed method on multiple datasets.

**Weaknesses:**

1. Some of the latest methods are not compared, such as RetinexFormer， and Rerinex-Diffusion. Besides, the experiments do not provide statistical indicators such as standard deviation or confidence interval in Table 1 and Table 2, which makes the reliability and stability of the results unclear. Moreover, the generalization ability of the proposed method is not explored, for example, when training the method in low-light environment and testing it one other exposure scenes.

2.  Although authors claim the proposed method is new, quantities of methods have explored introducing frequency-based techniques into image enhancement including some components-decomposition-based methods (such as [1])， what are the special characteristics of the proposed method that introducing the techniques that used above into retinex-based mechanism?

[1] Unveiling Advanced Frequency Disentanglement Paradigm for Low-Light Image Enhancement, ECCV 2024.

3.  There are too few visual results, for example, the visualization results of more real-world scenes (other low-light datasets or other exposure scenes) can be provided. Moreover, the detailed mechanism of why the proposed method works is not presented. How it has better performance than previous methods?

**Questions:**

Please see the Weaknesses.

---

> ### Author Response · Authors · 2024-11-18
>
> Thank you for your constructive feedback and recognition. Below are our responses, which we hope will address your concerns.
>
> **Answer2W1:**
>
> >Some of the latest methods are not compared, such as RetinexFormer, and Retinex-Diffusion.
>
> In our previous submission, given that our method is a zero-reference approach **designed to operate without ground truth data or normal-light data**, we focused on comparisons with the latest zero-reference methods. In light of your concerns, we have supplemented our experiments with comparisons against recent fully supervised methods. Since Retinex-Diffusion has not been open-sourced yet, we provide comparisons with another latest methods, RentinexMamba, on the LOL dataset. We will include relevant experiments as soon as Retinex-Diffusion becomes available.
>
> | Method        | LOLv1-PSNR(dB)$\uparrow$ | LOLv1-SSIM$\uparrow$ | LOLv2-PSNR(dB)$\uparrow$ | LOLv2-SSIM$\uparrow$ | SIDD-BRISQUE$\downarrow$ | SIDD-CLIPIQA$\downarrow$ |
> |---------------|----------------|------------|----------------|------------|--------------|--------------|
> | Retinexformer[1] | 23.93          | **0.831**      | 21.23          | 0.838      | 9.229        | 0.343        |
> | Retinexmamba[2]  | **24.03**          | 0.827      | **22.45**          | **0.844**      | 11.826       | 0.386        |
> | Ours          | 19.80          | 0.750      | 20.22          | 0.793      | **2.555**        | **0.292**        |
>
> The SIDD dataset is an unsupervised dataset that we constructed ourselves, so **these two supervised methods could not be trained on it**. We used the pre-trained model from LOLv1 for generalization testing. The results show that while supervised methods outperform zero-reference methods on paired datasets, **our method demonstrates superior generalization capability on non-paired (unlabeled) datasets**, achieving the best performance on the SIDD dataset.
>
> >Besides, the experiments do not provide statistical indicators such as standard deviation or confidence interval in Table 1 and Table 2, which makes the reliability and stability of the results unclear.
>
> Thank you very much for your suggestion. We fully agree with this point, as it helps to further validate the reliability of our results. However, some of the methods in our current comparison have not been open-sourced, and the corresponding original papers do not provide the necessary data, which makes a comprehensive and fair comparison impossible at this stage. We will supplement the experiments once the relevant works are open-sourced.
>
> >Moreover, the generalization ability of the proposed method is not explored, for example, when training the method in low-light environment and testing it one other exposure scenes.
>
> In light of your concerns, we explored the generalization capability of the proposed method on overexposed images. Specifically, we employed the model trained for 100 epochs on the LOL dataset to process overexposed images. The selected test dataset consists of the overexposed samples from the test set of the Exposure-Error Dataset in [3]. **You con find our visible results in the latest version of supplementary materials**.
>
> As illustrated in the experimental results **Fig.3**, the proposed method demonstrates generalization to overexposed scenarios and achieves accurate Retinex decomposition. At the same time, certain limitations are observed. Since our adaptive illumination adjustment module (LCNet) is trained on low-light datasets, domain discrepancies between the two tasks often result in lower output values from LCNet. This causes the enhanced image to maintain relatively high exposure levels. To address this issue, we followed enhancement measures of PairLIE[4], leveraging traditional techniques to adjust the decomposed illumination map. Specifically, dynamic range compression is employed. **The final output exhibits richer color gradations and exposure levels closer to human visual perception, compared to the input image**.
>
> [1] Cai, Yuanhao, et al. "Retinexformer: One-stage retinex-based transformer for low-light image enhancement." Proceedings of the IEEE/CVF International Conference on Computer Vision. 2023.
>
> [2] Bai, Jiesong, et al. "Retinexmamba: Retinex-based mamba for low-light image enhancement." arXiv preprint arXiv:2405.03349 (2024).
>
> [3] Afifi, Mahmoud, et al. "Learning multi-scale photo exposure correction." Proceedings of the IEEE/CVF Conference on Computer Vision and Pattern Recognition. 2021.
>
> [4] Fu, Zhenqi, et al. "Learning a simple low-light image enhancer from paired low-light instances." Proceedings of the IEEE/CVF conference on computer vision and pattern recognition. 2023.

---

> ### Author Response · Authors · 2024-11-18
> **reply to Reviewer SCTr**
>
> We sincerely appreciate the reviewer’s constructive feedback and recognition. We hope the following responses address your concerns effectively.
>
> >W2: what are the special characteristics of the proposed method that introducing the techniques that used above into retinex-based mechanism?
>
> **Answer2W2**: Thank you for providing the relevant recent literature, which has been very insightful for our ongoing research. We also noticed that the paper applies frequency modulation and decomposition methods in feature processing, with a particular focus on Laplace decomposition. The uniqueness of our frequency domain decomposition module lies in the following aspects:
>
> 1. **Prompt Learning Paradigm:** Our inspiration comes from the concept of prompt learning in generative models. Frequency domain decomposition places more emphasis on prior learning and utilizes a cross-attention mechanism to guide the retinal decomposition. This approach helps guide the network to retain frequency domain information as much as possible during L-R decomposition, avoiding direct feature concatenation operations that may affect the final optimization direction.
>
> 2. **Low-Frequency Concentration Property of DCT:** Unlike Laplace transforms, DCT focuses more on discrete signal processing, concentrating major information (such as textures) in lower frequencies. This helps separate high-frequency noise and non-salient information while preserving image details, making it highly suitable for our task.
>
> >W3:  the visualization results of more real-world scenes (other low-light datasets or other exposure scenes) can be provided
>
> **Answer2W3**: We have included additional experiments on the unsupervised low-light datasets LIME, NPE, MEF, DICM, and VV, **as discussed with Reviewer wCSY**. **Detailed results can be found in the latest supplementary materials**. Specifically, Fig. 2 in the supplementary materials demonstrates that our method outperforms other unsupervised methods in terms of generalization on the unsupervised datasets. Additionally, Fig. 3 in the supplementary materials shows that our method also exhibits a certain level of generalization on the overexposed dataset.
>
> >W4: the detailed mechanism of why the proposed method works is not presented
>
> **Answer2W4**: Apologies for any confusion caused. The inspiration for our method comes from two frameworks in the academic literature: unsupervised image denoising and unsupervised low-light enhancement. Unsupervised image denoising suggests that **two different noisy observations of the same scene can serve as mutual supervision, replacing the clean target signal**. This is explained in Section 3.1.2. On the other hand, unsupervised low-light enhancement proposes that **two different lighting observations of the same scene can be decomposed, with the reflection maps being equal**, and this is used as a loss function for gradient updates, as detailed in Section 2.1. Based on these, we naturally hypothesize that constructing image pairs with different lighting and noise observations can act as mutual supervision for collaborative enhancement and denoising. We adopted the downsampling method from self-supervised denoising, Neighbor2Neighbor, alongside traditional image dynamic range compression techniques to construct these image pairs.
>
> Additionally, we have provided a theoretical proof for this hypothesis, as discussed in Section 3.2. Through a Taylor expansion, we derive that the subimage after dynamic range compression can be decomposed into an enhanced illumination component and a reflection component with non-equivalent noise distributions. This can be formally expressed as:
> $\mathcal{D}_1(I)=(R_1 + N_1) \circ L_1, \overline{\mathcal{D}}_2(I)=(R_2+\lambda N_2)\circ \overline{L}_2$
>
> Therefore, we only need to **constrain the reflection R components of the two subimages after decomposition to be identical**, which **satisfies the assumptions of both self-supervised denoising and self-supervised enhancement**. We will include a more detailed explanation in the revised version of the paper. Thank you for your feedback.

---

### Official Review · Reviewer_Dq7D · 2024-11-04

**Soundness:** 3
**Presentation:** 3
**Contribution:** 3
**Rating:** 6
**Confidence:** 5

**Summary:**

This paper introduces an interpretable, unsupervised framework for joint denoising and low-light enhancement of images, focusing on real-world scenarios.

The method employs physical imaging principles and Retinex theory for decomposing images into illumination and reflection components.

**Strengths:**

1. The paper combined the Retinex model and data-driven methods since the first effort of RetinexNet.
2. Experiments are sufficient.
3. Frequency domain decomposition is well-used.
4. The unsupervised low-light enhancement method is a promising direction.
5.  The self-supervised denoising method based on neighboring pixel masking is well-aligned with the challenges of handling zero-reference images.

**Weaknesses:**

1. For low-light enhancement task, I want to see more results on LIME, NPE, MEF, DICM and VV. Since these datasets do not have ground truth, then is more fair to justify the effectiveness.

2. Missing citations of real-world low-light enhancement methods,
[1] Enhancing Visibility in Nighttime Haze Images Using Guided APSF and Gradient Adaptive Convolution

3. The paper lacks a discussion on the computational complexity and runtime efficiency of the proposed model.

**Questions:**

1. How does the proposed method handle edge cases such as extreme noise or heavy color distortions, which may not follow typical low-light degradation patterns?

2. Given the success of the DCT-based frequency decomposition, have the authors considered combining this with other frequency-domain transforms (e.g., wavelets) to enhance robustness across diverse types of degradation?

---

> ### Author Response · Authors · 2024-11-17
>
> We sincerely appreciate the reviewer’s constructive feedback and recognition. We hope the following responses address your concerns effectively.
>
> >W1: more results on LIME, NPE, MEF, DICM and VV
>
> **Answer2W1**: We have supplemented relevant experiments to verify the generalization capability of our algorithm, as **detailed in the discussion with reviewer wCSY**. **In comparisons with other state-of-the-art unsupervised low-light enhancement methods, our approach achieved superior performance**. Visualization results have been included in the revised version(Supplementary Materials: Figure 2, Tables 2 and 3).
>
> >W2: Missing citations of real-world low-light enhancement methods
>
> **Answer2W2**: Thank you for your suggestion. We have added comparison experiments with the latest methods in **the discussion with reviewer SCTr** and have provided a incorporation of the missing references in the **Introduction** section.
>
> >W3: lacks a discussion on the computational complexity and runtime efficiency of the proposed model
>
> **Answer2W3**: We fully agree with this perspective. Considering that previous work only compared parameter counts, we have supplemented the relevant tests. To ensure a fair comparison, all experiments were conducted on 128-pixel, three-channel images.
>
> | Method     | Flops(G) | Time(ms) |
> | ---------- | -------- | -------- |
> | Retinexnet | 14.23    | 7.37     |
> | LLFormer   | 3.46     | 51.99    |
> | SNR-aware        | 6.97     | 19.89    |
> | PairLIE    | 5.59     | 1.70     |
> | SCI        | **0.01**     | 1.52     |
> | Zero-DCE   | 1.30     | **1.39**     |
> | RUAS       | 0.05     | 3.72     |
> | Clip-LIT    | 4.56     | 2.10     |
> | Ours       | 5.10     | 16.56    |
>
> From the results, we find that our method is on par with some SOTA no-reference methods (e.g., PairLIE) in terms of FLOPs, but the inference time is relatively longer. Overall, compared to supervised methods (LLFormer, SNR-aware), our method still demonstrates higher computational efficiency. We analyze that the parts contributing to the increased computational load primarily include the frequency-domain decomposition using Discrete Cosine Transform (DCT) and the computation of cross-attention. This table have been included in the revised version(Supplementary Materials: Table 6).

---

> ### Author Response · Authors · 2024-11-17
>
> >Q1: How does the proposed method handle edge cases such as extreme noise or heavy color distortions?
>
> **Answer2Q1**: Our experiments on the SIDD dataset (please refer to Fig. 7 in main text) provide evidence of **the method's ability to generalize to real-world scenarios involving significant noise**. Additionally, the method exhibits **robustness after undergoing the sRGB camera pipeline transformation**.
>
> For low-light images with color distortion, we observed the limitations of existing zero-reference methods. In many scenarios, these models may struggle to correct images to a normal perceptual level. Compared to previous approaches, our model demonstrates greater robustness in handling these edge cases. We have included these edge cases in **sup.pdf, Figure 11**.
>
> The theoretical foundation for handling such challenging scenarios is grounded in the interpretable masking module, which adheres to noise principles, along with the color correction mechanism integrated into the loss function $L_{enh}$. These elements enable the model to effectively adapt to various types of degradation.
>
> >Q2: have the authors considered combining this with other frequency-domain transforms (e.g., wavelets) to enhance robustness?
>
> **Answer2Q2**:Thank you for your insightful suggestion. We appreciate your interest in enhancing the robustness of our approach by incorporating other frequency-domain transforms, such as wavelets. The motivation for using DCT lies in the explicit modeling and separation of complex degradations. We hope that such separation can serve as a guiding signal to assist in the optimization of the Retinex decomposition and strengthen the self-supervised learning process. Therefore, we adopted a relatively simple and direct approach involving masking and inverse transformation.
>
> While our method leverages DCT-based frequency decomposition due to its simplicity and effectiveness in separating multi-degradation components, we acknowledge that other transforms, such as wavelets, could potentially offer complementary benefits. Wavelets, for example, **allow for multi-resolution analysis, which might be advantageous for handling various levels of degradation across different scales**, which might not be fully captured by DCT alone. We have considered this possibility and plan to explore combining DCT with wavelet transforms in future work.
>
> Thanks for the reviewer's constructive feedback and recognition. If you have any further questions or concerns, please feel free to contact us.

---

### Author Response · Authors · 2024-11-21
**General Reply**

**Dear Committee Members and Reviewers,**

We are pleased to inform you that we have replied to all the reviewers' comments. and incorporated the following additional experiments into our revised submission:

1. **Generalization tests** on unsupervised datasets **LIME, NPE, MEF, DICM, and VV** (Supplementary Materials: **Figure 2, Tables 2 and 3**).
2. **Comparative experiments** on the **LSRW** dataset (Supplementary Materials: **Figure 5 and Table 8**).
3. **Generalization tests** on overexposed datasets (Supplementary Materials: **Figure 3**).
4. **Comparison of computational complexity and runtime efficiency** (Supplementary Materials: **Table 6**).
5. **Ablation study on the loss function** (Supplementary Materials: **Table 4**).
6. **Ablation study on different masking strategies and their ratios** (Supplementary Materials: **Table 7**).

The supplementary materials, detailed in the **sup.pdf** file, contain these additional results. Due to space constraints in the main manuscript and the associated effort required for reformatting, some of the critical findings have been included in the supplementary materials for your reference. We will incorporate these results into the main text in subsequent stages as needed.

We hope these additions address your concerns and provide further clarity regarding our work. Thank you for your time and constructive feedback. We look forward to your response and remain open to discussing any additional questions or suggestions.

---

### Meta-Review · Area_Chair_Kmpy · 2024-12-18

**Metareview:**

This paper proposes an interpretable unsupervised joint denoising and enhancement for real-world low-light scenarios. Experimental results show the effectiveness of the proposed method. All reviewers appreciate the contributions of the paper. However, they also raise lots of concerns including more evaluations results, discussions on the computational complexity and runtime efficiency of the proposed model, comparisons with state-of-the-art methods, e.g., RetinexFormer, and differences from eighbor2Neighbor and MAE.

The authors solve the concerns of reviewers. Based on the recommendations of reviewers, the paper is accepted.

**Additional Comments On Reviewer Discussion:**

During the discussion phase, Reviewer SCTr requests authors provide discussions of employing or extending it for other tasks. In addition, more qualitative results should be included as pointed out by Reviewer wCSY.

The authors provide more qualitative results in the rebuttal. All reviewers are satisfied with the rebuttal.

---

### Decision · Program_Chairs · 2025-01-22

Accept (Poster)